# PAM-repeat associations and spacer selection preferences in single and co-occurring CRISPR-Cas systems

Jochem N. A. Vink[1,2], Jan H. L. Baijens[1,2] and Stan J. J. Brouns[1,2*] 

* Correspondence: stanbrouns@gmail.com
[1]Department of Bionanoscience, Delft University of Technology, Delft, The Netherlands
[2]Kavli Institute of Nanoscience, Delft, The Netherlands

## Abstract

**Background:** The adaptive CRISPR-Cas immune system stores sequences from past invaders as spacers in CRISPR arrays and thereby provides direct evidence that links invaders to hosts. Mapping CRISPR spacers has revealed many aspects of CRISPR-Cas biology, including target requirements such as the protospacer adjacent motif (PAM). However, studies have so far been limited by a low number of mapped spacers in the database.

**Results:** By using vast metagenomic sequence databases, we map approximately one-third of more than 200,000 unique CRISPR spacers from a variety of microbes and derive a catalog of more than two hundred unique PAM sequences associated with specific CRISPR-Cas subtypes. These PAMs are further used to correctly assign the orientation of CRISPR arrays, revealing conserved patterns between the last nucleotides of the CRISPR repeat and PAM. We could also deduce CRISPR-Cas subtype-specific preferences for targeting either template or coding strand of open reading frames. While some DNA-targeting systems (type I-E and type II systems) prefer the template strand and avoid mRNA, other DNA- and RNA-targeting systems (types I-A and I-B and type III systems) prefer the coding strand and mRNA. In addition, we find large-scale evidence that both CRISPR-Cas adaptation machinery and CRISPR arrays are shared between different CRISPR-Cas systems. This could lead to simultaneous DNA and RNA targeting of invaders, which may be effective at combating mobile genetic invaders.

**Conclusions:** This study has broad implications for our understanding of how CRISPR-Cas systems work in a wide range of organisms for which only the genome sequence is known.

## Background

The adaptive CRISPR-Cas immune system provides heritable defense in the form of spacers, which are short nucleic acid sequences (28–36 bp) obtained from previous encounters with mobile genetic elements (MGEs). These are stored in the bacterial or archaeal chromosome in CRISPR arrays [1]. CRISPR arrays contain spacers flanked on both sides by repeat sequences (~ 30 bp) and are transcribed as

a single RNA, and subsequently processed into multiple crRNAs. crRNAs can be loaded into effector complexes formed by Cas proteins that subsequently scan the cell for nucleic acid targets. Base pairing between the spacer and target nucleic acids (protospacer) allows the specific binding of effector complexes to targets, which are then destroyed [2, 3]. CRISPR-Cas systems are widespread in bacteria and archaea, with 42% of bacterial and 85% of archaeal genomes containing a CRISPR-Cas system [4].

Both acquisition of new spacers (CRISPR adaptation) and target inactivation (CRISPR interference) are carried out by specialized sets of Cas proteins. *Cas* genes likely have originated from casposons [5], a family of self-replicating transposons, and have since evolved many new genes and gene variants [4]. Based on the evolutionary classification of their *cas* genes, there are two classes of CRISPR-Cas systems. Class I systems contain crRNA-effector complexes made up of multiple subunits, while effector complexes of class II systems are encoded by a single *cas* gene [4]. The two classes are further divided into six types, where each type is further divided into subtypes. The different types and subtypes do not occur homogeneously in nature, with class II systems being nearly exclusive to bacteria [4]. More than 95% of CRISPR-Cas systems found in complete genomes are one of the first three types: types I, II, or III [6].

CRISPR-Cas systems can be studied on a mechanistic or on a functional level. Mechanistic features describe how CRISPR-Cas systems are able to fulfill their role. The mechanisms through which they operate are diverse. For example, some CRISPR-Cas systems defend the cell by targeting DNA (e.g., types I, II, IV, and V), whereas other CRISPR-Cas types target invader RNA (e.g., type III and VI) [4]. Another important mechanistic feature is the presence of a protospacer adjacent motif (PAM), which DNA-targeting systems require to differentiate self from non-self [7–9]. Furthermore, the PAM is an important feature in the target search process of DNA-targeting systems within the cell [10, 11]. This motif sequence flanking the crRNA-pairing site, between one and five nucleotides long, not only differs between subtypes, but can also differ between *cas* gene orthologs within the same subtype, for example, Cas9 variants [12].

An important aspect of the PAM is the moment of selection. While a more stringent PAM selection is achieved during the adaptation stage by Cas1-Cas2 and in some systems Cas4 [13–15], PAM selection during the CRISPR interference phase by the crRNA-effector complex will also occur [16–18]. This led to the distinction of PAM into spacer acquisition motif (SAM) and target interference motif (TIM) [19]. In the above case where acquisition modules are more stringent, the PAMs that are observed are usually mostly determined by the acquisition machinery (SAM). However, in other situations, the observed patterns might have been the result of selection for a working TIM. For example, most of the spacers selected for in RNA targeting systems were found to be acquired at random [20], even though spacers present in natural CRISPR arrays often show a bias towards the coding strand [21, 22], suggesting that the bias emerged from effective interference spacers through natural selection. On the other hand, there are systems that contain a reverse transcriptase fused to Cas1 (RT-Cas1) [23] which can already select spacers from the correct strand. In experimental settings, these effects can be

separated, but in bio-informatic analyses of natural spacers, the resulting effect is a combination of acquisition selection and interference selection.

Functional features describe what purposes CRISPR-Cas systems fulfill within the cell. There is evidence for some CRISPR-Cas functioning beyond adaptive immunity [24, 25]; however, even within the context of an adaptive immune system, CRISPR-Cas systems can serve different roles (e.g., as the first line of defense or as an activator of other immune system pathways). This can be a reason why 23% of genomes with CRISPR-Cas systems contain more than one subtype [26], despite their costs [27, 28]. There are preferred combinations of certain subtypes, suggesting that there is an added benefit of having a specific combination of different subtypes present in the cell. The added benefit might consist of cooperativity between systems by the formation of different lines of defense, avoidance of type-specific CRISPR inhibition by MGE, or coupling of abortive infection mechanisms [26, 29–31]. On the other hand, some CRISPR-Cas systems are specialized to protect from certain invaders, which may require multiple co-occurring systems to be present in a single genome to protect from different types of invaders. Type IV systems that co-occur together with type I systems primarily target plasmids [32], and type III systems target a class of phages that other type I and V systems cannot [33, 34], indicating that specialization in targets is a potential reason for the co-occurrence of different subtypes. Through cooperation and specialization, co-occurring subtypes can function complementarily.

The functional and mechanistic features described above have been demonstrated experimentally for several microbial model systems, and these are often of specific interest to applications such as genome editing. High-throughput assays to identify the PAM of CRISPR-Cas systems have been developed but remain limited compared to the total range of CRISPR-Cas systems accessible bio-informatically [12, 35, 36]. The full diversity of PAM and other mechanistic and functional features of CRISPR-Cas systems in nature remain understudied. To improve our knowledge on the mechanistic and functional features of single and co-occurring CRISPR-Cas systems beyond the model organisms, we relied on vast metagenomic sequence databases to computationally find targets for spacers from diverse bacteria and archaea. This approach was recently taken to study phage-host interactions [37–39]. We mapped a third of the unique spacers to a target in publicly available metagenome sequence databases. We used the flanking regions of found spacer targets to build an initial PAM catalog of more than two hundred unique PAMs and for more than half of the spacers in CRISPRCasdb [6]. This was then employed to assign the correct orientation of transcription of CRISPR arrays, giving access to target strand information of invaders and uncovering conserved links between repeat ends and PAM. Through the quantification of the spacers targeting template or coding strands, we found that the preference for one of these strands is subtype-specific and indicates that some DNA-targeting systems (type I-E, type II-A, and type II-C) avoid RNA while other DNA- and RNA-targeting systems preferentially target RNA (type I-A, type I-B, and type III systems). We found spacers in co-occurring CRISPR-Cas systems to be compatible with both PAM and strand requirements, indicating that they may be shared between systems and will lead to both DNA and RNA targeting. Lastly, we identified three categories of multi-effector compatible spacers, which meet the PAM and strand requirements of co-occurring DNA and RNA-targeting systems.

## Results

### Blast analysis finds matches for 32% of spacers from CRISPRCasdb

The first step in our analysis was to select a set of CRISPR spacers and find potential matches to these sequences in DNA sequence databases. To this end, we selected the previously described CRISPRCasdb, which contained all spacers from 4266 complete bacterial and archaeal genomes [6]. The spacers from CRISPRCasdb were then mapped to the sequences from the NCBI nucleotide database as well as metagenomic databases with a high number of prokaryotic and virus sequences. Matches between spacers and sequences from the databases were found using BLASTn [40]. The matches were then filtered using an optimized approach which increased the number of matches while keeping the false positives to a minimum (the "Methods" section, Additional file 1: Fig. S1A). As an indication of the false-positive rate, we determined that for the matches found in the NCBI nucleotide database, 1% were eukaryotic or eukaryotic viral sequences, 10% were prokaryotic viral sequences, and the majority (88%) corresponded to prokaryotic genome sequences (Additional file 1: Fig. S1A). This specificity towards prokaryotic sequences in a database that contains predominantly (83%) eukaryotic sequences shows that even though false-positive hits cannot be excluded, the false-positive rate is low.

From the 221,850 total unique spacers analyzed, this optimized filtering approach resulted in 72,099 spacers (32% of total) with at least one match (Fig. 1A), of which 31,327 spacers (15% of total) had a match in the NCBI nucleotide database (Fig. 1B). For more than 25,000 of these, the best hit was completely identical to the spacer, and for the vast majority (60,294), the total number of mismatched nucleotides was three or less (Fig. 1C). Also, in most cases, more than one hit was found per spacer (Fig. 1D).

The fraction of spacers with matches differed greatly between different genera, with *Streptococcus*, *Pseudomonas*, and *Staphylococcus* among the genera with the highest fraction of matches (77%, 69%, and 64%, respectively) and *Calothrix*, *Nostoc*, and *Thermosipho* among the lowest (4%, 4%, and 3%, respectively) (Fig. 1E). The genera with high spacer matches typically occurred in well-sampled environments (human-associated), whereas the genera with lower matches occurred in what appear to be poorly sampled environments (soil, oceanic). A previous study [41] which looked for spacer matches in the NCBI nucleotide database found matches for 7% of spacers, using a more stringent 95% sequence identity and 95% coverage cutoff as filtering thresholds. This difference in the fraction of spacers with matches in the NCBI nucleotide database indicates the added benefit and importance of our more sensitive filtering process. Additionally, the number of sequences in the database has increased in recent years from ~ 230 billion to ~ 700 billion bases. The most important factor for the increase in the number of spacers with matches however was the use of metagenomic databases, as the majority of unique spacer matches derived from these databases (Fig. 1B, Additional file 1: Fig. S1B).

To find the subtypes of the spacers, we aligned the CRISPR repeat sequences to repeat sequences with known subtypes, based on the method described by Bernheim et al. [26]. Except for subtype II-B for which we extracted 453 spacers, all analyzed subtypes from type I, II, and III systems contained more than a thousand spacers (Fig. 1F). An exceptionally high fraction of spacers with matches was found for subtypes II-A

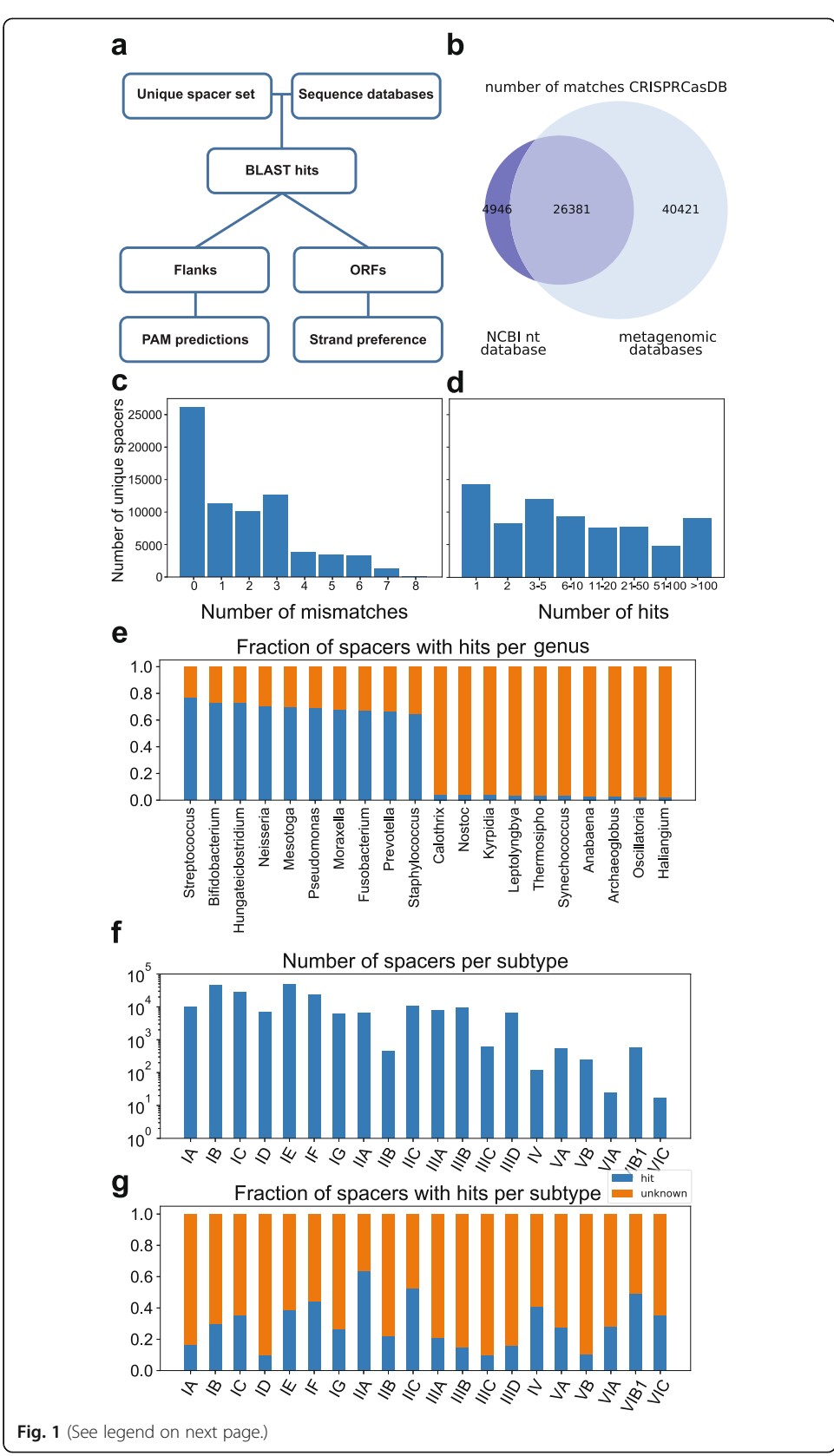

**Fig. 1** (See legend on next page.)

(See figure on previous page.)
**Fig. 1** Spacer targets found with BLAST. **A** Computational pipeline for finding spacer targets. Targets of 72,099 spacers were found using blastn and filtered based on the fraction of spacer nucleotides matching a target sequence (see the "Methods" section). **B** Venn diagram of spacers with matches in the NCBI nucleotide database vs metagenomic databases. **C** Plotted is the number of unique spacers (total 72,099) for which a match was found. Generally, spacers < 4 mismatches fall within > 90% identity threshold and are selected directly, and spacers with 4 or more mismatches generally within the > 80% and < 90% threshold and were selected in case another spacer from the same genus targeted the same sequence. **D** Number of sequences targeted by each spacer. Due to redundancy in the datasets, some of these sequences can be identical. **E** Fraction of spacers with hits for the ten genera with the highest and ten genera with the lowest fraction of hits. Only genera with at least 500 spacers are shown. **F** Number of spacers per subtype. The subtype of a spacer was predicted based on the similarity of the repeat sequence to repeats with a known subtype (see the "Methods" section). **G** Fraction of spacers with hits per subtype

(63%) and II-C (53%), while subtype I-A, subtype I-D, and type III subtypes had notably lower fractions of spacer matches than average (15%, 11%, and 20%, respectively; Fig. 1G). The differences in fractions of matches found between subtypes may be due to their phylogenetic distributions, where well-sampled genera have different subtypes than poorly sampled genera (see above). However, even within well-sampled genera, the fraction of spacers with matches differs between subtypes, with type III subtypes having fewer hits on average (22%) than other subtypes (38%). The biases that we observed for both the fraction of hits in certain genera and subtypes remained true when we only used perfectly matching spacers (Additional file 1: Fig. S2). Overall, the large number of spacers with matches revealed sets of sequences that were targeted by each CRISPR-Cas subtype, which were then used to study mechanistic and functional aspects of CRISPR-Cas defense.

### Alignment of protospacer flanks reveals 220 unique subtype-specific PAMs covering 55% of spacers

One of the important mechanistic features of CRISPR-Cas defense for DNA targeting systems (types I, II, IV, and VI) is PAM recognition [8, 19, 42, 43]. The first PAM was discovered in the alignment of bacteriophage sequences that were targeted by *Streptococcus* spacers [44]. Later studies revealed more PAMs or the effect of mutant versions of the PAM [16, 45–47]. We expand on these known PAMs that are limited to well-studied organisms by predicting new PAMs based on the alignment of the flanks of spacer matches (protospacers). The potential of this method for large-scale PAM predictions was shown in a previous bioinformatics study [48], with a key limiting factor being the number of spacers with matching targets. It was also previously shown that PAMs, acquisition machinery, and repeat clusters co-evolve [19]. We therefore increased the number of spacers with matches within one group by clustering spacers based on repeat similarity (> 90% nucleotide identity and same repeat length). The sensitivity of PAM detection depends on the information content of the nucleotide positions of the PAM (signal) compared to the information content of the other flanking positions (noise). We found that clustering based on repeat similarity increased the signal-to-noise ratio for PAM detection compared to clustering based on species-subtype (e.g., *Escherichia coli* I-E) or genus-subtype (e.g., *Pseudomonas* I-F). We furthermore found that spacers originating from organisms with very high or low GC contents displayed

increased noise. We thus further increased the signal-to-noise ratio by adjusting the expected frequency of flanking nucleotides based on the average GC content of the spacers within the cluster (Additional file 1: Fig. S3A). The flanks of unique hits within each cluster can subsequently be aligned, and with enough spacer hits, the information content reliably reveals the PAM sequence and position relative to the protospacer (Fig. 2). We further checked whether our filtering approach leads to optimal PAM prediction and found that with stricter hit requirements (95–100% identity), the signal-noise ratio of PAM prediction decreased (Additional file 1: Fig.

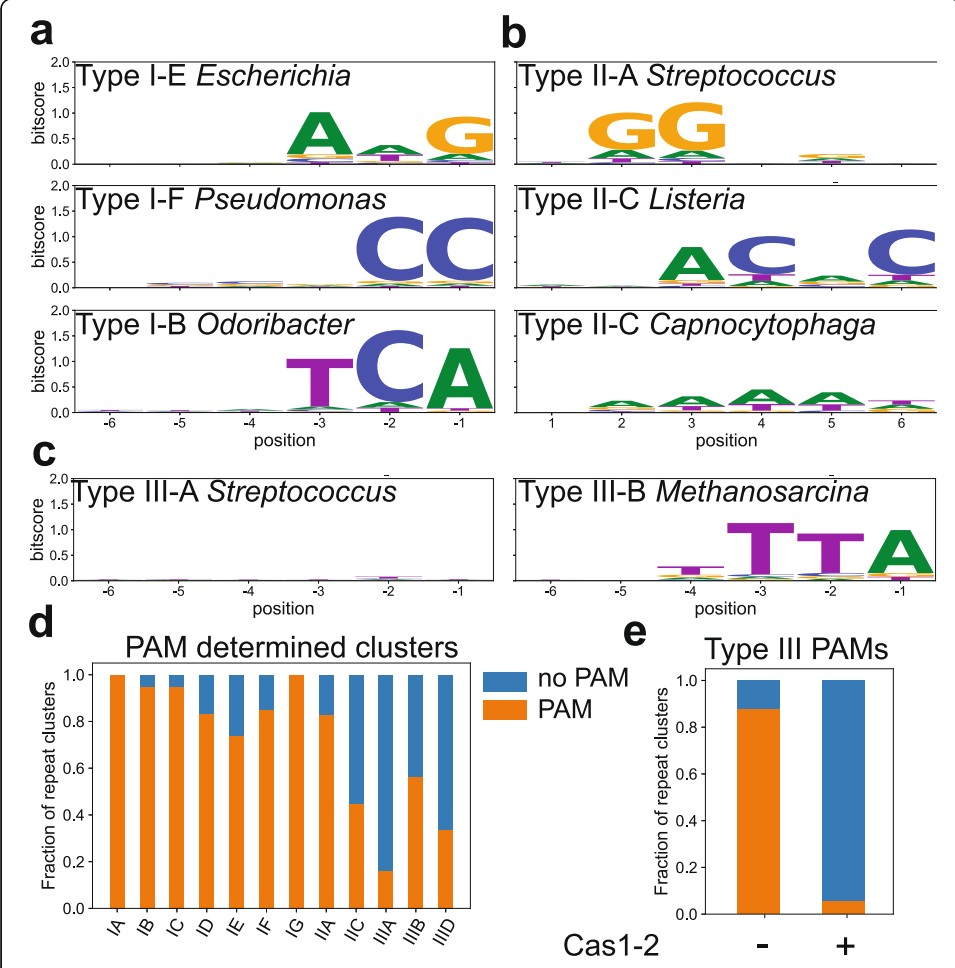

**Fig. 2** PAM determination of repeat clusters. **A** Sequence logos of upstream flank of hits to spacers from type I repeat clusters. Sequence logos of protospacer flanking regions per repeat cluster. *Y*-axes show the information content per nucleotide position. The label includes the subtype of the repeat cluster and a representative genus in which this repeat cluster is found. The PAM of the I-E and I-F repeat clusters depicted here has been determined previously in model systems containing the same repeat [49, 50]. **B** The same as **A** but for downstream flanks of spacers from type II repeat clusters. The PAM of the type II-A (*Streptococcus*) and type II-C (*Listeria*) systems has been previously described in model systems that are closely related to the strains studied here [51, 52]. **C** The same as **A** but for upstream flanks from type III repeat clusters. **D** Frequency of PAM-determined repeat clusters with more than 25 hits. Nucleotide positions were considered part of PAM with a bitscore of at least 0.4 and 10 times above the median bitscore of the 23 nucleotides surrounding the hits. PAM size was at least 2 nucleotides. **E** Frequency of PAM-determined repeat clusters for type III systems that contain Cas1-2 vs type III systems that lack Cas1-2. Additional file 5 contains the PAM for each strain-subtype combination (Additional file 2)

S3B). This was caused by a lower number of spacers per cluster and the number of hits per spacer.

This clustering approach together with our large number of hits led to a PAM prediction for 123,144 spacers (55% of all spacers; Additional files 2 and 3). For type I and type IV, the PAM is known to occur in the 5′ (upstream) flank of the protospacer, while type II systems have their PAM in the 3′ (downstream) flank of the protospacer [1] (Fig. 2A). This well-characterized feature of the PAM therefore allows the unique possibility to correctly orient CRISPR arrays given the rules described above. The orientation of arrays is an important feature to properly identify the chronology of acquisition events, the CRISPR leader sequence and potential RNA targeting. Tools have been developed to predict these bio-informatically [53–55]. However, these tools in some cases contradict each other, implying that this prediction is not straightforward and fully accurate [54, 56].

To measure the accuracy of CRISPR array orientation predictions, we compared the predictions to experimentally determined the orientations from a recent study using transcriptome sequencing (TOP) to determine the direction of transcription of arrays [56]. The 7968 experimentally inferred spacer orientations were the same as our predictions in 85% of cases, while only 33% of TOP predicted spacer orientations were the same as the CRISPRCasdb prediction (Additional file 2) which is a combination of CRISPRdirection and a GC content-based leader prediction tool [55, 57]. For the 15% where TOP did not match our predictions, both CRISPRCasdb and our PAM-based orientations predicted the same orientation, indicating that some of the TOP orientation predictions based on transcription data might not have been correct. When we compared the predictions of CRISPRCasdb with our PAM-based orientations directly, we found a 88% match between all spacers. We furthermore found that many type I and type III repeats for which we predicted the orientation based on the PAM contained the 3′-end motif ATTGAAAC of their repeat (Additional file 1: Fig. S4) described previously [58]. This conserved motif is transcribed and forms the 5′ handle of the crRNA and is held by crRNA-effector complexes. Altogether, these findings indicate that the position of the PAM is a reliable indicator for the orientation of the CRISPR array, and can be used to annotate CRISPR array information, giving access to features such as spacer acquisition chronology and strandedness.

### Type I PAMs are conserved within repeat types, and type II PAMs are strain-specific

Sequence logos of alignments of type I (Fig. 2A) recover previously known PAMs including the subtype I-E AWG PAM found in *Escherichia* and subtype I-F CC PAM found in *Pseudomonas* [59], but also previously undescribed PAMs. Out of the 43 unique PAM-subtype combinations, 25 were not found in previous publications (Table 1). Interesting examples of novel PAMs include a CTT PAM in I-C systems (compared to the more canonical TTC) and a CCA PAM in I-F systems (compared to the more canonical CC). They are generally short (2–3 nt) and are well-defined (high information content/bit score). Diversity is highest in I-B systems (11 unique PAMs) and lowest in I-F systems (3 unique PAMs).

For type II PAMs, we found both short, well-defined PAM motifs (such as *Streptococcus* II-A) as well as longer PAMs with less conserved PAM motifs (Fig. 2B). Poorly

**Table 1** Unique type I PAM sequences. Table of all unique type I PAMs found for the different subtypes and representative genera that contain the repeat cluster for which each PAM was determined. For previously described PAMs, a reference ID has been added which correspond to the following: 1 [60], 2 [61], 3 [62], 4 [63], 5 [45], 6 [64], 7 [65], 8 [46], 9 [14], 10 [66], 11 [67], 12 [49], 13 [68], 14 [69], and 15 [70]

| Type | PAM | Genus | Ref |
|---|---|---|---|
| **I-A** | ATG | *Leptospira* | **1** |
| | CCN | *Acidianus* | 2 |
| | TTA | *Thermodesulfobacterium* | |
| | TCN | *Sulfurisphaera* | 3 |
| | ATN | *Aminobacterium* | |
| **I-B** | CCA | *Moorella* | **4** |
| | CCN | *Clostridium* | 4 |
| | CCT | *Ureibacillus* | 4 |
| | TAC | *Halorubrum* | |
| | TCA | *Campylobacter* | |
| | TCN | *Campylobacter* | |
| | TTA | *Methanosarcina* | **5** |
| | TTC | *Halobacterium* | **6** |
| | TTN | *Novibacillus* | **7** |
| | TTTA | *Petrimonas* | |
| | TTG | *Thermobacillus* | |
| **I-C** | CTN | *Anaerobutyricum* | |
| | CCN | *Porphyromonas* | |
| | CTT | *Ruminococcus* | |
| | TTC | *Geobacillus* | 8 |
| | TTN | *Acidovorax* | |
| | TTT | *Lachnoclostridium* | |
| **I-D** | GCN | *Haloquadratum* | |
| | GGTG | *Halorubrum* | |
| | GTN | *Methanotrix* | 9 |
| | GTT | *Microcystis* | |
| | GTG | *Methanospirillum* | 10 |
| **I-E** | AAC | *Bifidobacterium* | 11 |
| | AAG | *Geobacter* | 12 |
| | AAN | *Klebsiella* | |
| | AAT | *Lactobacillus* | |
| | AAA | *Kosakonia* | 13 |
| | AC | *Corynebacterium* | |
| | AG | *Xenorhabdus* | |
| | AWG | *Escherichia* | |
| **I-F** | ACC | *Aeromonas* | |
| | CC | *Pseudomonas* | 14 |
| | CCA | *Pseudomonas* | |
| **I-G** | TAC | *Rothia* | |
| | TAN | *Propionibacterium* | 11 |
| | TTN | *Pseudopropionibacterium* | 15 |
| | AAN | *Acidipropionibacterium* | |
| | TTC | *Rhodothermus* | |

conserved PAM motifs could be caused by a variation of PAMs used within the same repeat cluster or by the promiscuity of PAM recognition in type II systems [71]. In previous work, it was shown that in some cases, Cas9 proteins that use the same repeat can have different PAMs [12, 72]. We questioned whether our clustering of spacer hits based on repeat sequence would result in the low conservation scores in some PAM motifs. When we based our PAM motif predictions on spacers coming from a single genome, we recovered different PAMs for type II systems that use the same repeat (Additional file 1: Fig. S5A-D), whereas for type I systems, we always recovered the same PAM for each genome within a repeat cluster. We conclude that repeat sequence clustering is not an option and therefore only useful to derive PAMs from spacers in type II systems in individual genomes. From this genome-based clustering of spacers, unique PAM sequences were recovered from 302 genomes in type II systems. We could verify that all experimentally determined PAMs matched our predicted PAMs, suggesting that the predictions were reliable.

Overall, the diversity in PAM motifs in type II systems is higher than in type I systems (Additional file 1: Fig. S5E-F). For type I, we found hits for 56,026 spacers, from 588 different genera in 34 different phyla. For type II systems, we found hits for 9883 spacers from 149 different genera in 14 different phyla. Based on these numbers, you would expect the number of unique type I PAMs we recovered to be higher than type II PAMs. However, in total, we find 43 unique PAM-subtype combinations for type I compared to 134 unique PAM-subtype combinations for type II systems. Of these, type II-C is the most diverse, which matches earlier studies showing the structural and sequence diversity of type II-C Cas9s [12, 73]. Rapid shuffling of the PAM-interacting domains (PID) of Cas9s could drive the extended PAM diversity in type II systems [74].

### Forty-three percent of type III repeat clusters contain a PAM

When investigating type III repeat clusters, we found many devoid of a PAM. This is expected, as RNA-targeting systems do not require a PAM to find a target (Fig. 2C), and rely on the protospacer flanking sequence (PFS) to avoid self-targeting [75, 76]. Interestingly, other repeat clusters contained PAMs that appeared to be the same as type I PAMs, which raised the question, why these clusters contained a PAM. We compared the PAM detection frequency for clusters with at least 25 unique spacer hits (Fig. 2D). For type I subtypes whereas for type III systems, the number of PAM-containing repeat clusters was lower, with type III-A having the lowest (16%) and III-B the highest (56%) fraction of PAM-containing repeat clusters in type III systems. As it was previously shown that type III systems often lack their own acquisition machinery [77], we hypothesized that the PAM found in type III repeat clusters originates from the spacer acquisition machinery that type I systems share with type III systems. We observed that the PAM frequency in type III clusters that lack their own acquisition machinery is high (95%; Fig. 2E), whereas the PAM frequency is low in type III clusters that contain their own *cas1-cas2* genes (8%). This supports the hypothesis that the PAM in type III arrays originates from type I spacer acquisition modules functioning in *trans*. Genomes with PAM-containing type III systems can be found in Additional file 5.

### Conserved patterns between PAM and repeats

PAMs usually differ from the ends of CRISPR repeats, which allows for self-nonself discrimination [8, 46, 78]. Type III and other RNA-targeting CRISPR-Cas systems do not require a PAM, but many do require mismatching between the repeat end and the protospacer flanking sequence (PFS) [79, 80]. Given these previous observations, we wanted to investigate if there are conserved links between repeat ends and PAM of individual systems (Fig. 3A) and whether type III PAMs that originate from type I spacer acquisition modules are also compatible with type III PFS requirements.

We collected all unique repeat-PAM sequence combinations in our dataset and compared the repeat nucleotide with the corresponding PAM nucleotide in each position. For type I systems (Fig. 3B), we found that the − 3 and − 2 nucleotide of the repeat can be a strong predictor of the corresponding PAM nucleotide, where a − 3C in the repeat would lead to a − 3A in the PAM, − 3G to − 3T, − 3T to − 3A. At the middle position, a − 2C would lead to a − 2A in the PAM (Fig. 3B). The most common − 2 and − 3 repeat nucleotide is an A, in which case the PAM nucleotide mostly is either a T or a C. For the − 1 position, the nucleotide identity of the PAM sequence cannot be predicted directly from the repeat sequence.

For type II systems, most nucleotide positions can accommodate two or three PAM nucleotides (Additional file 1: Fig. S6A). In + 2 and + 3 positions, the most common repeat nucleotide (T) accommodates either an A or G PAM nucleotide, which is analogous to the most common nucleotide in type I systems (− 3 and − 2 adenine), which tends to co-occur with a C or T PAM nucleotide. For type III systems, the variation of

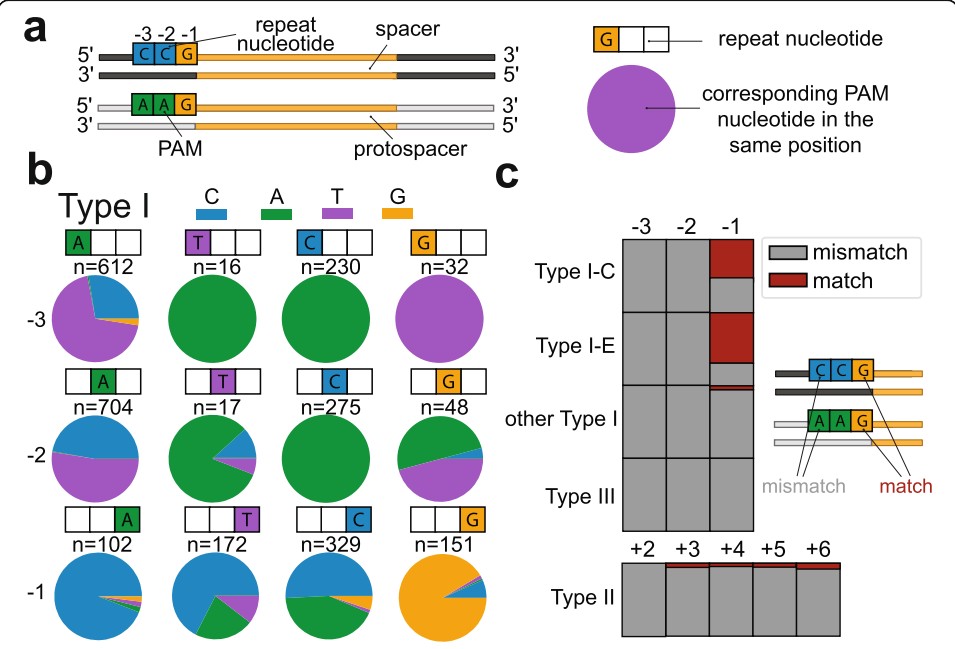

**Fig. 3** Relationship between repeat and PAM sequence. **A** Schematic of the analysis of PAM and repeat sequence. The nucleotide identity of the PAM in each position is compared to the nucleotide of the repeat. **B** PAM nucleotide frequency for type I repeats. For each given repeat nucleotide position (indicated with colored boxes), the PAM nucleotide (pie chart) for each unique PAM-repeat combination of our database. The number of occurrences is indicated above the pie chart (*n*). **C** The frequency of matches (red) and mismatches (gray) between the PAM and the corresponding repeat nucleotide for each position in relation to the spacer. For type II, the positions are compared on the other side of the spacer

repeat nucleotides is smaller, but generally similar combinations are found as in type I systems (Additional file 1: Fig. S6B). Overall, the most conserved repeat-PAM co-occurrence patterns are found in the − 2 and − 3 positions of the type I and type III arrays.

These co-occurrence patterns suggest that in most cases, the PAM that is used and selected differs from the repeat. This holds true for most of the experimentally determined and previously predicted PAM sequences [8, 14, 61, 62, 70, 81, 82]. However, previous studies have shown that in some cases, part of the repeat sequence is PAM-derived [49]. We then asked in what CRISPR-Cas subtypes the PAM matches the corresponding repeat nucleotide for each of the spacer flanking positions. When we counted the occurrence of a matching PAM, we found that this only occurred frequently in the − 1 position of type I-C (35%) and type I-E (48%; Fig. 3C). We found that these matches are associated with repeats that have TTC PAMs in type I-C and AAG PAMs in type I-E, which could indicate that the C of type I-C repeat sequences is PAM-derived, as was similarly demonstrated for the G of AAG PAMs in type I-E [49].

In other positions and CRISPR-Cas types, > 98% of the repeat-PAM combinations did not match each other, which shows that the general patterns between repeats and PAMs, and perhaps the mechanism of self- vs non-self discrimination are conserved in all subtypes. In type III systems, all cases demonstrate mismatches between PAM and repeat, which is a requirement of functional type III spacers [79, 80]. This finding demonstrates that the PAMs of type III array spacers acquired with type I acquisition modules are compatible with PFS requirements of type III systems.

### Strand bias for the template or coding strand is subtype-specific

Our method has revealed a large number of newly identified PAMs and has shown that type III systems which lack their own acquisition machinery and co-occur with type I systems almost always contain a PAM. The presence of a PAM in these systems could enable type I systems to use the spacers stored in type III arrays as they are compatible with the PAM requirements of type I effector complexes. Furthermore, type III effector complexes could benefit from a PAM-selecting acquisition module, as it excludes spacers with repeat-PAM matches (Fig. 3C).

Besides the PFS, another requirement for type III spacers is that the spacer comes from the correct strand, as these complexes can only bind to the RNA transcripts. We wondered whether some species indeed use type I and III dual functionality CRISPR arrays, as PAM-dependent DNA targeting and PAM-independent mRNA targeting are not mutually exclusive. We therefore asked whether spacers of DNA-targeting systems are also compatible with type III surveillance complexes, if they happened to be picked from the correct strand.

To determine the potential ability of crRNA to target RNA, we measured the strand bias by counting the spacers that targeted the coding or template strand of predicted open reading frames (ORFs) (Fig. 4A). As spacers targeting the template strand are unable to base pair the transcribed RNA, the fraction of spacers targeting the coding strand serves as an estimate of the RNA targeting ability of the crRNA. For example, in *Moraxella* IIIB arrays, a significant bias for the coding strand was found (88%, $p < e^{-11}$) (Fig. 4B). This bias allows type III effectors carrying crRNA from those spacers to bind

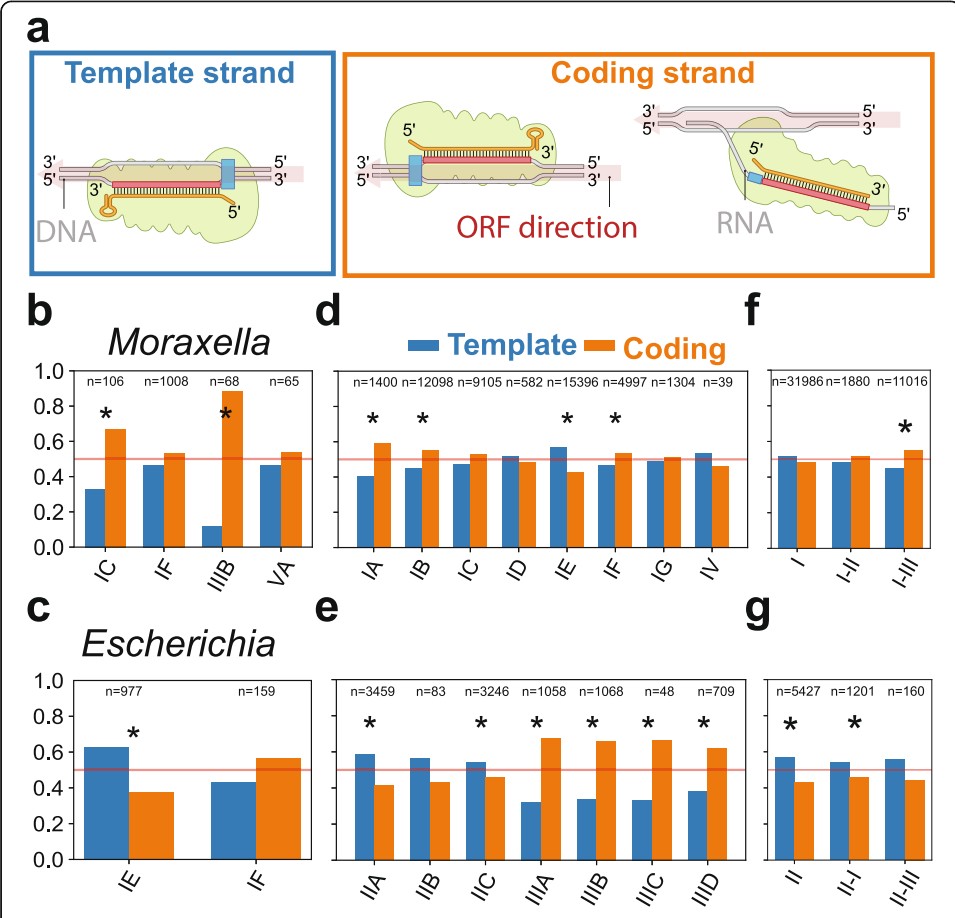

**Fig. 4** Template and coding strand targeting of spacers. **A** Schematic representation of a spacer targeting the template strand and a spacer targeting the coding strand inside an ORF. Spacers targeting the coding strand are also able to base pair with and target transcribed RNA. **B** Fraction of *Escherichia* spacers targeting template (blue) and coding (orange) strand by subtype. **C** Fraction of *Moraxella* spacers targeting template and coding strand by subtype. **D** Fraction of spacers targeting template and coding strand for type I and type IV subtypes. **E** Fraction of spacers targeting template and coding strand for type II and type III subtypes. **F** Fraction of spacers targeting template and coding strand for type I. Spacers are grouped based on which other types of Cas effector genes are present in the genome. **G** The same as **F** but for type II spacers. The significance of strand bias is calculated with a binomial test, and a *p*-value < 0.01 is indicated with an asterisk. Additional file 2 contains the strand targeted of each spacer and allows to extract the strand bias for each taxon

to their target RNA. However, also I-C spacers in *Moraxella*, for whose effectors this is not strictly required, show significant bias for the coding strand ($p < e^{-3}$), indicating a selection for RNA-targeting spacers.

For *Escherichia* subtype I-E, 977 spacer matches inside ORFs were found, of which 611 (63%) targeted the template strand (Fig. 4C), showing a significant bias for targeting the template strand ($p < e^{-14}$) potentially avoiding RNA. No significant strand bias was found for *Escherichia* subtype I-F (43% template strand, $p = 0.11$), suggesting that strand bias is CRISPR-Cas subtype-specific.

Analysis of our complete dataset revealed general trends in the strand preferences for each subtype (Fig. 4D, E). The strongest strand bias was found in type III systems with an average of 65 of the spacers matching the coding strand (coding strand: template strand ~ 2:1). This result demonstrates that there is selection in type III systems for

spacers to target the transcribed RNA. This selection can originate at the adaptation stage by dedicated adaptation machinery selecting from RNA/coding strands such as RT-Cas1 [23] or at the interference stage, where only functional RNA-targeting spacers are retained in the population [20].

The strand biases we found are consistent with our curated CRISPR array orientation predictions, because an incorrect CRISPR array orientation prediction would obscure strand-specific targeting. Type I-A and type I-B also displayed significant strand bias for the coding strand although at lower levels (60% and 55%; $p < e^{-9}$ and $p < e^{-14}$, respectively).

Contrary to the type III, type I-A, and I-B systems, we found a significant strand bias towards the template strand in subtype I-E, type IV, and type II systems, with the strongest bias found in subtype II-A (59%) and subtype I-E (57%). Given the high number of spacers in these groups, the chance of observing this bias by chance is small ($p < e^{-23}$ and $p < e^{-69}$, respectively), again suggesting avoidance of RNA.

### Co-occurrence of type I and type III systems lead to PAM and strand targeting compatibility

As we noticed that type III spacers were compatible with type I PAMs in multiple cases, we next asked whether type I spacers are compatible with RNA targeting in microbes with co-occurring type I and III systems. We measured the strand bias of type I spacers in genomes containing either a combination of type I, type II, and type III surveillance complexes (Fig. 4F). No significant strand bias was found for type I spacers in the presence of type I and/or type II surveillance complexes. However, in the presence of type I and type III surveillance complexes, type I spacers had a slight but significant coding strand bias (55%, $p < e^{-14}$). This might be caused by increased selection pressure to keep RNA targeting spacers in the presence of RNA targeting surveillance complexes. This would suggest that spacers are selected to be compatible for both type I and type III effector complexes in such situations. For type II spacers, the presence of type III did not significantly change the strand bias (Fig. 4G). Given the natural tendency of type II spacers to bias towards the template strand (Fig. 4E), these findings suggest that type II spacers are less compatible with co-occurring type III effector complexes than type I spacers.

### Three distinct categories of co-occurring multi-effector compatible arrays exist

The findings above indicate that subtype-specific preferences exist for either the template or coding strand of the DNA. These preferences might enable or preclude compatibility between the spacers of co-occurring subtypes. The subtype-specific preference of template strand targeting (e.g., in type I-E and type II) will reduce the number of effective spacers that can be used by co-occurring RNA-targeting systems, whereas subtypes with a preference for the coding strand (type I-A, type I-B) might make their spacers more compatible with RNA-targeting systems. We categorized all multi-effector compatible arrays that can be used by effector complexes from different subtypes. This means for co-occurring DNA-targeting systems, these arrays need to have a PAM that can be used in both systems, whereas for the co-occurrence of a

DNA target CRISPR-Cas system with an RNA targeting system, the arrays present in the genome need to both have the correct PAM and have a bias for the coding strand.

Overall, we can distinguish three main categories of co-occurring CRISPR-Cas systems in which spacers are compatible for multiple effectors (Fig. 5A, Additional file 4): firstly, two co-occurring DNA-targeting systems which have their own adaptation machinery and their own repeat sequences (Fig. 5B; *n* = 7; type I-A–type I-B, 5; type I-B–type I-C, 2); secondly, a co-occurring DNA-targeting and RNA-targeting system, with distinct repeat sequences but a commonly shared acquisition machinery (Fig. 5C; *n* = 17; type I-B–type III, 15; type I-E–type III, 3); and thirdly, a co-occurring DNA-targeting and RNA-targeting system, with shared repeat sequences and shared acquisition machinery (Fig. 5D; *n* = 85; type I-B–type III, 71; type I-A–type III, 11; type I-C–type III 3).

Taken together, our data indicate that multi-effector compatible arrays are most prevalent between type I and type III systems. Within the type I systems, the most

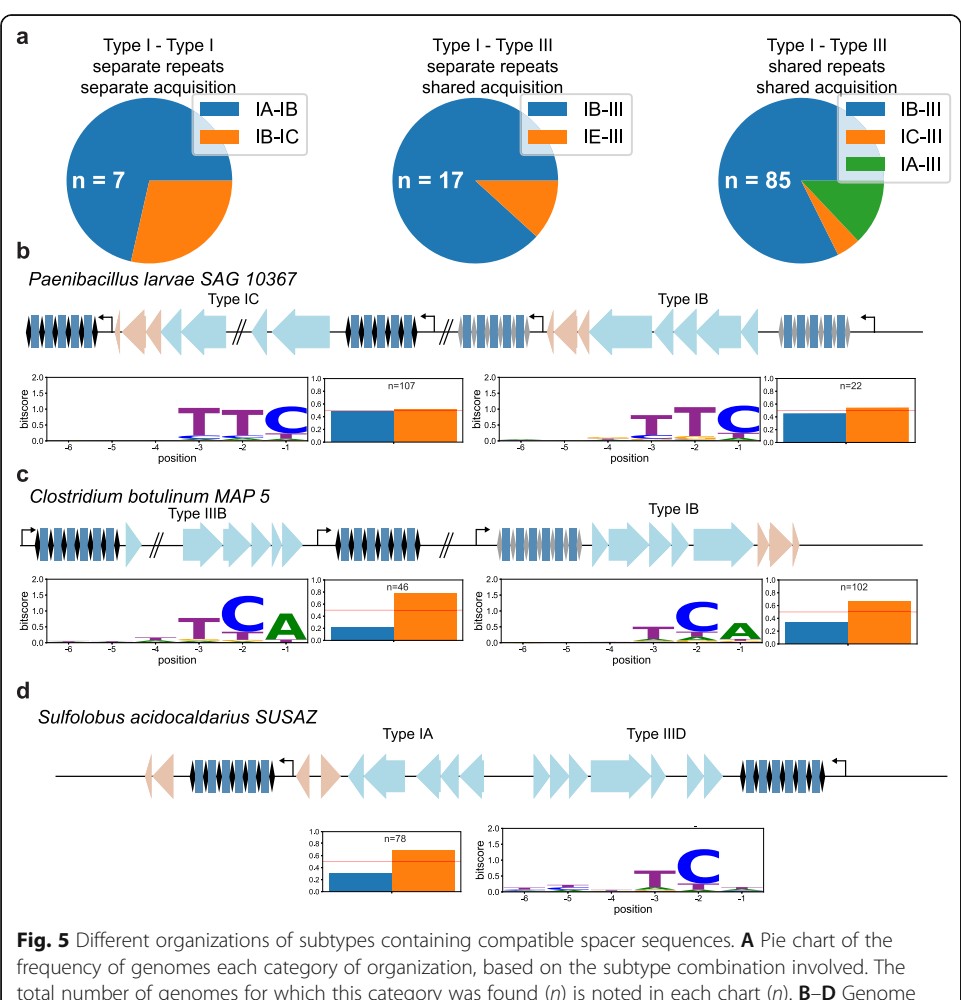

**Fig. 5** Different organizations of subtypes containing compatible spacer sequences. **A** Pie chart of the frequency of genomes each category of organization, based on the subtype combination involved. The total number of genomes for which this category was found (*n*) is noted in each chart (*n*). **B–D** Genome representations of the examples for the different organization categories. **B** Type I-type I compatibility. **C** Type I-type III compatibility (different repeat sequences). **D** Type I-type III compatibility (same repeat sequences). Genes involved in interference (blue) and adaptation (red) are shown for the different subtypes within the genome. PAM logo and strand bias of each associated repeat cluster is depicted below the genomic representations

common subtype to use multi-effector compatible arrays is type I-B, but also type I-A, type I-C, and type I-E use these arrays. The type III systems that use compatible arrays lack their own adaptation machinery; however, repeat clusters in these co-occurring systems display a strand bias that suggests selection for RNA-targeting spacers. The information content is similarly strong for PAMs in type III arrays as in type I arrays, which demonstrates that the PAM is selected to the same extent for type I as shared type III arrays.

## Discussion

In this study, we have matched CRISPR spacers of complete genomes of bacteria and archaea with their targets in (meta)genome databases and subsequently analyzed the genomic flanks of the protospacers. We computationally found targets for 32% of CRISPR spacers from thousands of bacterial and archaeal genomes. This is a major increase in spacer targets compared to previous studies and is due to our sensitive filtering process and use of metagenomic databases [41]. We found that type III spacers had the highest fraction of unknown targets of any CRISPR-Cas type. This was not solely caused by the phylogenetic or environmental occurrence of type III systems, because the fraction of type III spacers with unknown targets within a genus was typically higher than that of other types. This means that the targets of type III systems are either under-sampled or that type III spacers contain more mismatches to their targets, making them harder to find computationally. Recently, a single new study doubled the number of known RNA viruses including phages [83], while another study greatly increased the number of known single-stranded RNA phages [84], indicating that RNA phages have been poorly sampled. We predict the fraction of spacers with matches to increase with increasing numbers of available metagenomic data, especially including more RNA viruses and more data from poorly sampled environments.

By analyzing the flanks of the spacer hits in great depth, we have generated a catalog of PAM sequences for each CRISPR repeat cluster. The repeat sequence is a good predictor of parts of the PAM sequence in type I and outperformed clustering based on genus-subtype classifications. This finding is corroborated by the position-wise comparison of PAM and repeat nucleotides, which shows certain repeat nucleotides predict PAM nucleotides. This may be helpful to either predict the PAM from scratch, or to further experimentally determine the PAM while reducing the degeneracy at certain positions, limiting the predicted PAM sequence space. However, for type II systems, this repeat-based PAM prediction does not work, because PAM motifs are not conserved within each repeat sequence cluster. Instead, PAMs in type II systems seem to be conserved within a CRISPR-Cas system combined with a certain repeat in individual strains (Additional file 1: Fig. S5C-D). Strain-specific PAM analysis in type II systems uncovered a large diversity of PAM sequences, much larger than the PAM diversity in type I systems. Further analysis could perhaps base clustering on the PAM-interacting domains of Cas9 protein sequences, which might serve as a better predictor for PAM sequence conservation than the repeat sequence [12, 74].

The mismatch between repeat and PAM nucleotides generally holds, except for the type I-E and type I-C, where for some repeat clusters the repeat nucleotide matches the PAM at the − 1 position. The most common PAMs of these systems (TTC for I-C; AAG for I-E) are also complementary to each other. These findings indicate type I-C

systems could have a similar mechanism of spacer acquisition with a PAM-derived last repeat nucleotide as in type I-E [49], even though these systems do not share related Cas1 proteins [85] or repeat structures [58]. The crystal structures of the Cas1-Cas2 adaptation machinery from both systems [86, 87] indicate that the same strand is probed for the PAM (5′GAA3′ in I-C and 5′CTT3′ in I-E), which demonstrates that this phenomenon has not arisen from complementary strand selection.

The PAM catalog can be used to predict the PAM for arrays in newly sequenced genomes and metagenomic contigs in type I and type III systems if they contain repeats that are closely related to the repeats in our database. We furthermore have uncovered novel PAMs in type II systems, and together, these developments give access to unexplored mechanistic and biotechnological potential. For repeats that are not in our database, the nucleotide identities of the repeat in the spacer flanking positions can be used to predict, with lesser certainty, which PAM it could have and select certain CRISPR-Cas systems of interest for further study.

Furthermore, the position of the PAM in the target is a reliable indicator for the orientation of transcription of CRISPR arrays. Correct prediction of transcription of CRISPR arrays gives access to measuring chronology of invader encounters and strand-specific targeting of CRISPR-Cas systems, which is especially relevant for RNA targeting CRISPR-Cas. The spacers of type III systems, which target RNA, have a bias towards targeting coding strands, making them capable of base pairing and thereby targeting RNA. Unexpectedly, we also found several subtypes with a preference for the template strand (I-E and type II). The reason for this type of strand bias is not yet clear, but we pose that this could be caused by a selection for spacers that do not target RNA (RNA avoidance), as DNA targeting with these spacers might be impacted by inactivating complementary RNA [88]. In addition, there might be a difference in binding or dislodging of crRNA effector complexes from the template strand vs coding strand by RNA polymerase [10, 89]. Lastly, we cannot exclude the possibility that DNA replication might cause the observed strand bias for some subtypes, as transcription and replication are often co-oriented in prokaryotes, plasmids, and phages [90, 91].

We have categorized multi-effector compatible CRISPR arrays whether they share the same repeats and/or acquisition machinery and whether only DNA or both DNA and RNA are targeted. DNA-targeting systems that use multi-effector compatible arrays generally have their own acquisition machinery, and the low frequency of this co-occurrence in nature might indicate that this is not actively selected for. It needs to be experimentally verified whether the spacers in these compatible arrays are actually shared between complexes. However, some crRNA sharing between DNA systems has already been observed experimentally, so it is therefore likely to be found for more systems [92].

Multi-effector compatible arrays are much more common in co-occurring DNA- and RNA-targeting systems. The strand bias that occurs in type I arrays indicates that type III effector complexes are using these spacers and thereby creating selection pressure on the RNA binding potential of the transcribed crRNA. It also seems that the most commonly co-occurring type I systems (I-A, I-B, and I-C) that use compatible arrays also have the largest coding strand bias. Whether this strand bias is induced by the presence of type III or whether these subtypes by their nature have a strand preference and therefore became more commonly compatible with type III systems is not yet clear.

Interestingly, many of the subtype combinations that share PAMs also co-occur more often than expected by chance, suggesting they have positive epistatic interactions [26]. Furthermore, repeat sequences of type I-A and I-B are in the same repeat families as type III repeats, providing further indications of their compatibility [58].

The experimentally determined spacer sharing in *Marinomonas mediterranea* [30] described previously does not fall within the categories in this study as the type III system has its own adaptation machinery. In this case, the systems are not mutually compatible because the type I systems cannot use the type III spacers due to a lack of PAM, which we have not further investigated in this study. Also, the other previously experimentally described spacer sharing systems in *Pyrococcus* [92] and *Flavobacterium* [31] were not found due to a lack of sufficient hits, which demonstrates that this bioinformatic analysis likely underestimates the number of systems that can cooperate.

The discovery of multi-effector spacer compatibility in a large number of genomes in this study together with previous experimental evidence of spacer sharing of RNA and DNA-targeting systems [30, 76, 92] shows that there is selection pressure to share spacers cooperatively within arrays. The evolutionary benefits of such cooperativity could be profound. Firstly, as two subtypes generally have different mismatch tolerance [93–95], targeting the same sequence with two subtypes can reduce the probability of escape mutation. Secondly, a combination of RNA and DNA targeting systems can provide multiple layers of defense, where RNA targeting might give more time for DNA-targeting systems to destroy the invader before the cell is taken over [10]. Thirdly, the length of arrays in a genome has recently been shown to be limited by auto-immunity [96]. By sharing spacers, each subtype is supplied with a maximum diversity of spacers while self-targeting costs are minimized. Lastly, the different mechanisms these systems use allow for complementary and distinct benefits. The priming mechanism [97, 98], unique to DNA-targeting systems, can accelerate spacer acquisition for both systems, whereas the cOA signaling pathways [99, 100], unique to type III, could activate defense systems that benefit both systems.

## Conclusion

Altogether, this study highlights the wealth of information that can be retrieved by analyzing the targets of CRISPR spacers on a large scale. It furthermore demonstrates under what conditions CRISPR-Cas systems can cooperate and provides a catalog of PAM predictions and targeted MGEs awaiting further study.

## Methods

### CRISPR spacers and sequence data

A total of 221,089 spacers along with information on *cas* gene presence, genome, and repeat sequence were obtained from CRISPRCasdb [6] in February 2020, and the taxonomy of the genomes was obtained from the NCBI Taxonomy database [101]. We created our own sequence database by combining all sequences from the NCBI nucleotide database [102, 103], environmental nucleotide database [104], PHASTER [105], Mgnify [106], IMG/M [107], IMG/Vr [108], HuVirDb [37], HMP database [109], and data from Pasolli et al. [110]. All databases were accessed in February 2020.

Subtypes were predicted based on the repeat sequences using the subtype predictions and method described by Bernheim et al. [26], where the subtype of a spacer was inferred by the similarity of its repeat sequence to repeat sequences with known subtype (74% identity threshold to infer subtype).

### Blast hits and filtering

Hits between spacers and sequences from the aforementioned databases were obtained using the command line blastn program [40] version 2.10.0, which was run with parameters word_size 10, gapopen 10, penalty 1, and an *e*-value cutoff of 1, to find as many potential targets as possible. These blast hits were then filtered to remove hits of spacers inside CRISPR arrays and false-positive hits found by chance. Hits inside CRISPR arrays were detected by aligning the repeat sequence of the spacer to the flanking regions of the spacer hit (23 nucleotides on both sides). This alignment was done using the globalxs function from the Biopython pairwise2 package [111] with – 3 gap open and – 3 gap extend parameters. If more than 13 nucleotides were identical in the alignment of at least one flank, the hit was suspected to fall inside a CRISPR array and was filtered out.

To minimize the number of hits found by chance, we filtered hits based on the fraction of spacer nucleotides that hit the target sequence, as this metric considers both the sequence identity and the coverage of the spacer by the blast hit. In the first step, only hits with this fraction higher than 90% were kept. To find targets for even more spacers while keeping the number of false positives low, we included a second step where hits with a fraction higher than 80% were kept if another spacer from the same genus hits the same contig or genome in the first step. This second step did not introduce hits on any new contigs or genomes and was based on the assumption that multiple spacers from the same genus hitting the same contig or genome is unlikely to be caused by chance. Finally, we removed spacers that were shorter than 27 nucleotides (54 spacers) and removed 7 spacers that were hitting aspecifically, such as inside ribosomal RNAs or tRNAs. This left 72,099 unique spacers with target hits for downstream analysis.

### Protospacer flank alignment for orientation and PAM predictions

The PAM is known to occur on the 5′ end of the protospacer for type I, type IV, and type V CRISPR-Cas systems and on the 3′ end for type II systems [1, 112]. We used this property to predict the orientation of transcription of CRISPR arrays and the sequence of crRNA. The PAM sides were compared to the nucleotide conservation in the flanking regions of the spacer hits, and the spacer orientations were predicted such that the flank with the greater conservation matched the known PAM side.

To measure the nucleotide conservation in the flanking regions, data from multiple spacers was combined based on the subtype and repeat sequences of the spacers. Highly similar repeat sequences from the same subtype were clustered using CD-HIT [113] with a 90% identity threshold. We hypothesized that similar repeat sequences would be used in a similar orientation and would utilize the same PAM sequences, as coevolution of PAM, repeat, and Cas1 and Cas2 sequences has been shown previously [58, 114]. For each repeat cluster, the flanking regions of the spacer hits were aligned. To equally weigh each spacer within the repeat cluster, irrespective of the number of

blast hits, consensus flanks were obtained per spacer. These consensus flanks contained the most frequent nucleotide per position of the flanking regions. From the alignment of consensus flanks, the nucleotide conservation, or information content, in each flank was calculated in bitscore [115] using the Sequence logo python package. We corrected for GC content of the targeted sequences by calculating the expected occurrences of each nucleotide based on the GC content of the spacer sequences. To minimize the number of orientation predictions based on little or noisy data, we only predicted the orientation for repeat clusters when the alignment of consensus flanks consisted of at least 10 unique protospacers. Furthermore, the information content of at least two positions was higher than 0.3 bitscore and higher than 5 times the median bitscore calculated from 23-nt flanks on both sides. These parameters were chosen as strictly as possible, while still yielding orientation predictions for the highest number of spacers.

Using the orientation predictions described above, we predicted the PAMs for each repeat cluster by checking which nucleotide positions were conserved. To minimize PAM predictions based on noise, we only predicted the PAM for repeat clusters where the alignment of consensus flanks consisted of at least 10 unique protospacers. A nucleotide position was predicted to be part of the PAM when higher than 0.5 bitscore and higher than 10 times the median bitscore. These parameters were chosen as strictly as possible, while maximizing the number of repeat clusters with PAM predictions and minimizing the number of unique PAMs predicted.

We subsequently categorized and counted multi-effector compatible spacers in the following ways: firstly, by an occurrence of multiple repeat clusters with different subtype classification that both contained the same PAM, either two DNA targeting clusters (category I) or a DNA and an RNA targeting cluster (category II); secondly, if multiple *cas* gene clusters from different subtypes were in the vicinity of a single repeat cluster and their genomes did not further contain other arrays linked to these *cas* gene clusters they were counted as a third category multi-effector compatible array.

### Coding vs template strand targeting analysis

For each spacer target inside an open reading frame (ORF), we determined if the spacer targets the coding (DNA and RNA) or template strand (DNA-only) during transcription. The ORFs and their orientation were predicted using Prodigal [116] for one target sequence per spacer. The target sequence of each spacer was selected as the longest hit sequence in the NCBI nucleotide database, excluding "other sequences," or, if no such sequence was hit, the longest hit sequence in metagenomics database. Using our spacer orientation predictions for type I, II, and IV spacers, and the orientation predictions from CRISPRCasdb for the other spacers, we checked if the spacer target (blast hit orientation) was on the coding or template strand of the predicted ORF. To test for significant bias towards either the temperate or the coding strand, a two-sided tailed binomial test was performed with an expected probability of 0.5.

### Supplementary Information

---

**Additional file 1.** Supplementary figures.

**Additional file 2** CSV file containing for each unique spacer in the CRISPRCasDB the following columns: **Spacers:** spacer sequence; **Repeats:** repeat sequence in host(s) (can be multiple if multiple genomes contain same spacer)**;**

**Accessionnrs:** accession number of host(s)**; Subtype:** subtype of array**; cas_genes**: Cas_genes present in host(s)**; hit**: if match found in (meta)genomic database equals 1 (else 0)**; consensus**_flanks: consensus sequence of left and right flank from flanks of all the hits in databases to this spacer**; repeat_cluster:** id of repeat cluster generated with CD-hit**; strandbias:** Orientation of hit in reference to ORF (1 coding strand 0 template strand, -1 undetermined)**; type:** Type of CRISPR array**; orientation_CRISPRcasdb:** Orientation of spacer determined in CRISPRCasDB [6]**; orientation_PAMbased:** Orientation of spacer determined in this study based on PAM**; orientation_TOPbased:** Orientation of spacer determined with TOP [56]**; PAM:** PAM sequence of repeat cluster (if predicted)**; Genus, Family, Order ….:** Taxonomy of host**; Type I, TypeII, Type III…:** Whether host genomes contain genes related to specific Type (1 yes, 0 no)**; Subtypesingenomes:** Which subtypes are in genomes**; Subtypesinproximity:** Which subtypes are in proximity (<25000 bp from spacer)**; Proximity_subtypes:** Distance of spacer to gene cluster of specific subtype**; subtypesCas1**: Which subtypes are in genomes that contain a Cas1 protein.

**Additional file 3.** CSV containing PAM catalog (each unique repeat for which PAM was determined) with following columns: repeat, PAM and subtype.

**Additional file 4.** CSV containing genomes for which compatible arrays were found with following columns: accession number genome, compatible subtypes of array, PAM, category.

**Additional file 5.** CSV containing genomes for which PAM was predicted with following columns: PAM, accession number, subtype.

**Additional file 6.** Review history.

### Acknowledgements
The authors thank Christine Pourcel and Pierre-Albert Charbit for supplying the CRISPRCasdb in a spacer-based format and all members of the Brouns groups for the input during group discussions.

### Peer review information

### Review history
The review history is available as Additional file 6.

### Authors' contributions
S.B. and J.V. conceived and supervised the project. J.V. gathered the databases. J.V. and J.B. wrote the analysis scripts. J.V., J.B., and S.B. wrote the manuscript. The author(s) read and approved the final manuscript.

### Funding
S.B. is supported by the Netherlands Organisation for Scientific Research (NWO VICI; VI.C.182.027) and has received funding from the European Research Council (ERC) CoG under the European Union's Horizon 2020 research and innovation program (grant agreement No. [101003229]).

### Availability of data and materials
The datasets on which the analysis is based have been submitted as additional files. Scripts to reproduce figures are available on request.

## Declarations

### Ethics approval and consent to participate
Not applicable

### Consent for publication
Not applicable

### Competing interests
The authors declare that they have no competing interests.

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

## 