## [**Additional file 6.** Review history. · Genome Biology]

Review History

First round of review

Reviewer 1

Are you able to assess all statistics in the manuscript, including the appropriateness of statistical tests used? Yes, and I have assessed the statistics in my report.

Comments to author:

Vink et al. present a study of a CRISPR spacer diversity and sequence analysis and provide insights into patterns of origin, targeting and function using a series of in silico analyses that lead to bioinformatically-supported hypotheses and interpretations. The overall goal and scale of the analyses are noteworthy and timely (with current tools and datasets enabling such analyses in ways previously less accessible and feasible), and some of the study findings are interesting and even intriguing. The overall topic is also timely and of interest to an increasing readership, though some of the foci are perhaps of interest to a limited subset of bona fide CRISPR experts.

The manuscript narrative overall suffers from a disjointed series of analyses, interpretations, arguments and discussions that do not seem to share a common thread, which in combination with niche foci and a specialized and relatively highly technical level of details makes the overall paper a hard read for the non-CRISPR expert, and perhaps even some of the CRISPR readership, let alone a broad readership. This is apparent in the title (I am not convinced that "comprehensive PAM prediction" is the anchor of the study), which is hard to decipher and interpret and may not fully represent what has been done and what is most interesting in the paper (I am also not convinced that "spacer sharing" is a highlight of the study). This also applies to the abstract, though it is more clear and articulate. I think variants of "large scale analysis of CRISPR spacers reveals sampling biases and provides mechanistic insights in immune function" could and perhaps should be considered. As a side note, but an important technical aspect, PAM sequences and their predictions are independent from spacer sharing, and mechanistically agnostic to strand biases, so even experts could find this somewhat confusing.

Though many quantitative details and numbers are provided, I think some key aspects are missing and the authors must provide some information on materials and methods, as well as results, to provide a more thorough assessment of the work and interpretation of the data. With 200k spacers total and 70k encompassed in the study results, it is important to understand how many of these were perfect matches, or almost so (only 1 SNP) and how many were "distant matches". Providing a bar graph showing the numbers of spacers vs. level of identity or #/% of mismatches (SNPs and gaps) is likely necessary and could enable the authors to dig deeper in the best matches to have even more conclusive interpretations. On that note, in some cases, the authors seem to provide less "supported" arguments that focus on very specific and small subsets of the data (e.g. only 114 PAMs deciphered from such a large dataset seems implausible and much of the text is occasionally focused on a small subset of hand-picked cases (see later comment). With regards to materials and methods, the authors should discuss length and percent identity requirements and cutoffs for the blast hits and filtering and consider a deeper analyses of the best hits. With over 70k unique spacers making the cut, there should surely be

some intriguing findings to glean from the set with perfect or near-perfect matches. I also imagine that there is a correlation between match quality and the ability to predict PAMs, which the authors should investigate. I also wonder what the distribution of these perfect matches is (e.g. supplementary figure 1) and what the proportion of perfect matches to prokaryotic viruses and also plasmids are.

As the authors know, some studies have investigated in depth naïve adaptation and primed adaptation in select organisms with known and well-characterized CRISPR-Cas systems (e.g. Marraffini, Severinov, Barrangou, Moineau and others). In cases where results are noteworthy (e.g. strand biases), one wonders whether these studies confirm and support these conclusions. To a similar extent, this applies to recent studies providing large scale analyses of the CRISPR spacers to the virome (e.g. Dion et al., 2021; Camarillo-Guerrero et al., and perhaps others, presenting highly relevant tools and data). This also applies to the relationship between CRISPR repeats and PAM sequences, with a need for the authors to determine whether this is indeed the case in a number of well-documented CRISPR-Cas systems with known PAMs. In many instances, the authors should ensure they properly discuss the highly relevant literature. It can admittedly be challenging to keep up with the CRISPR literature, but some of these references are highly relevant to provide context here.

In some cases, the authors show subsets of data that seems highly relevant and interesting (e.g. high matches for *Streptococcus*, *Pseudomonas* and *Staphylococcus*), and later on data with high levels of spacers with hits in select genera (see figure 1E, with perhaps the top 4-5 genera warranting deep analyses). Perhaps these should be analyzed a bit more in depth and given at least as much attention as select examples that otherwise stand out. In other cases, the authors seem to focus on relatively small numbers of cases to make strong conclusions (e.g. 7 genomes, 17 genomes and 85 genomes for the three co-occurring systems compatibility discussion).

The 114 PAMs covering approximately 55% of spacers is intriguing and seems like a relatively low number of PAMs for such a large number of systems and spacers. Can the authors determine how many of these PAMs are known and already characterized (perhaps half of them have already been characterized and possibly most of them already predicted?).

The comments and discussion about the critical use of the PAM for CRISPR array annotation and orientation is important. The authors should use this to discuss more extensively the quantitative extent to which previous studies have mis-predicted the orientation of CRISPR arrays.

Another critical and intriguing finding is the presence of PAMs in 43% of Type III systems, which is in disagreement with mechanistic insights and the literature and could prompt a series of studies by the authors and perhaps others. This may need more discussion, highlighting and perhaps discussing specific type III systems that should be subjected to experimental analyses. Since cas genes working in trans has been shown before (e.g. with split arrays that share repeats and machinery), it is not unprecedented nor unbelievable that some Cas machinery could be shared between CRISPR-Cas systems, but the implications for DNA-targeting type I systems and RNA-targeting type III systems is quite noteworthy and warrants additional coverage in my opinion. I actually wonder whether the authors could determine how often (or not) spacers are

shared between these systems, and whether there are patterns of "activity" that could be investigated (are the array sizes similar reflecting "equivalent" evolutionary acquisition patterns between the two loci sharing acquisition machinery, or are there noteworthy differences that seem to indicate that the acquisition machinery is favoring one locus/system/CRISPR array, and is it in cis or in trans in these cases?). Perhaps some deep dive into the systems discuss in figure 5 is warranted.

While the targeting strand bias is noteworthy, and has been discussed in the past in the literature, the authors must more clearly state that this is a biased and not a binary dataset. There are indeed biases (e.g 55%, 63%, 65%, ratios of approximately 2:1) and they do seem convincingly statistically significant, but they do not seem to have been selected against and there are explanation (e.g. steric access limitations due to other molecular mechanisms such as transcription) that could explain biases. Perhaps the authors could use transcriptomic data to determine whether transcription level correlates (or not) with biases.

Editorial comments:

- In the context of a system, the authors should and perhaps must use CRISPR-Cas (CRISPR and cas genes, and CRISPR and Cas proteins), as opposed to just "CRISPR", which refers to the repeat-spacer array. CRISPR itself and without Cas/cas is not a system...
- In some cases, especially when discussing the original PAM discovery and naming convention, the authors should and perhaps must cite the 2008 papers by Deveau et al. and Horvath et al. which observed and documented and showed the actual PAM. The preceding Bolotin et al. 2005 paper did note a bias but predicted a perplexingly incorrect PAM and the subsequent 2009 Mojica paper renamed a previously established CRISPR motif. Given how familiar some of the authors are with the historical literature, it could be useful and valuable to the readership to cite some of the original literature (2007-2010), though the more recently published reviews cited are certainly defensible. This also applies to some recently published and highly relevant papers (self vs non-self; CRISPR array identification tools, PAM identification and visualization tools and perhaps more like the two aforementioned virome-CRISPR spacer matching studies).
- The authors should consider using the original verbiage and words used by the authors (e.g. CRISPRdb, CRISPRCasdb with the lower case database). In some instances, the authors use CRISPRCasDB, CRISPRCasDb and perhaps other variants.
- In some cases, the authors should consider toning down some quantitative claims (e.g. 114 PAMs with perhaps as many already documented in the literature is not a "vast catalog", nor a "large number")

Reviewer 2

Are you able to assess all statistics in the manuscript, including the appropriateness of statistical tests used? No, I do not feel adequately qualified to assess the statistics.

Comments to author:

In this work, Vink and coworkers perform extensive bioinformatics analyses on CRISPR spacers, gaining insights into subtype-specific PAM preferences, biases in target strand selection, and potential crosstalk between co-occurring systems. Multiple studies have analyzed spacer

sequences broadly, although the limited number of matches have restrained any major insights. The authors report an incredible increase in matches—presumably due to the increasing accumulation of metagenomic sequences, allowing them to garner these insights. While many of the insights the authors draw are known and understood in the field (e.g. subtype-specific PAM preferences, PAMs and repeats being strongly different), the authors do provide additional insights as well as a large-scale analysis that confirms what the field had observed anecdotally. My comments noted below should strengthen the work and make it a stronger resource for the CRISPR community.

Major comments:

1. Because the work relies solely on bioinformatics analyses, some of the conclusions need to be softened or expanded with other explanations. As one example, on line 335 - 337, the authors state that the co-occurring systems may use each other's spacers. However, a competing explanation is that the acquisition machinery recognizes both repeats, but differences in the repeats prevents spacer sharing. Providing experimental evidence would be even stronger, although I recognize this would add considerable work.
2. The identified PAMs reflect requirements for spacer acquisition, while the requirements for targeting can deviate strongly and tend to be much more relaxed (e.g. acceptance of PAMs by *E. coli* Cascade outside of AAG). However, the authors often focus on targeting when assessing PAMs. This distinction between acquisition and targeting requirements could be clearer throughout, and I recommend that the authors start from the perspective of acquisition sequences that lead to targeting.
3. PAMs have been experimentally determined for a large number of Type II systems (e.g. Gasiunas et al. Nat Comm 2020). Incorporating these known PAMs into some of the analyses would provide more experimental support. It is also worth noting that PAMs can vary between highly similar Cas9 nucleases with similar repeats, complicating analysis of PAMs associated with clusters.
4. Could the direction of replication affect the bias in protospacer orientation? Such biases are known in bacterial genomes. If they exist in viral or plasmid genomes as well, then that could explain biases—particularly based on the typical source of spacers from one subtype to the next. This bias could upend the authors' assertions about preferring or avoiding RNA targets.

Minor comments:

5. The writing was generally clear, although there were typos and grammatical errors scattered throughout the text that were distracting.
6. Line 36: I would recommend "CRISPR immunity" rather than "CRISPR interference", as the latter is commonly understood to mean programmable gene repression with CRISPR.
7. Line 59: a much more recent review on alternative functions was by Wimmer & Beisel,

Front Microbiol 2020.

8. Line 77 - 78: This is an overstatement and should consider the use of cell-free transcription-translation systems (e.g. Marshall et al. Mol Cell 2018) that are easier than the IVT system cited by the author. Instead, the valid argument is that experimental systems require testing one system at a time that can't cover the same space as bioinformatics approaches.
9. Line 85 - 87: the logic seems circular here, where PAM predictions were used to improve PAM predictions.
10. Line 98: better define what is meant by a match. Is this a perfect match? Or with some mismatches? This is likely covered in the methods, although this should be noted here given that identifying spacer matches is the root of the entire work. Also, does their definition help explain why the authors find many more matches than prior attempts at spacer matching?
11. Line 196: is there evidence of distinct sets of PAMs within these clusters? This is worth closer inspection.
12. Line 197: add Dugar et al. Mol Cell 2018 and Rousseau et al. Mol Cell 2018, which also showed RNA targeting by the CjeCas9 and NmeCas9 without any PAM requirements.
13. L. 184: this is a bold claim, especially since none of the provided examples seem particularly novel compared to what's been experimentally determined (particularly for Cas9 nucleases).
14. Line 260 - 300: while the authors do observe significant biases based on binomial distributions, the differences are often minor by eye. These smaller differences should be reflected in the text, such as by providing actual percentages throughout.
15. Line 318 - 319: It's not obvious at this point in the text has any preferences would enable or preclude compatibility. Perhaps add another sentence or two.
16. Line 329 - 331: spacer sharing would be extremely unlikely given recognition of a hairpin and handle for the I-B system and pairing with the tracrRNA for the II-A system. The more plausible explanation is that these systems recognize common PAMs that happen to be the complement of each other.
17. Line 345: a nice experimental example to work in the story was an example of one set of acquisition machinery incorporating spacers into a co-occurring II-C and VI-B system (Hoikkala et al. mBio 2021).
18. Line 378 - 381: Or could the I-C system recognize an A-rich sequence on the opposite strand? Checking available crystal structures could address this possibility.
19. Line 407 - 410: This sentence could be shortened or split in two.

Reviewer reports:

Reviewer #1: Vink et al. present a study of a CRISPR spacer diversity and sequence analysis and provide insights into patterns of origin, targeting and function using a series of in silico analyses that lead to bioinformatically-supported hypotheses and interpretations. The overall goal and scale of the analyses are noteworthy and timely (with current tools and datasets enabling such analyses in ways previously less accessible and feasible), and some of the study findings are interesting and even intriguing. The overall topic is also timely and of interest to an increasing readership, though some of the foci are perhaps of interest to a limited subset of bona fide CRISPR experts.

The manuscript narrative overall suffers from a disjointed series of analyses, interpretations, arguments and discussions that do not seem to share a common thread, which in combination with niche foci and a specialized and relatively highly technical level of details makes the overall paper a hard read for the non-CRISPR expert, and perhaps even some of the CRISPR readership, let alone a broad readership. This is apparent in the title (I am not convinced that "comprehensive PAM prediction" is the anchor of the study), which is hard to decipher and interpret and may not fully represent what has been done and what is most interesting in the paper (I am also not convinced that "spacer sharing" is a highlight of the study). This also applies to the abstract, though it is more clear and articulate. I think variants of "large scale analysis of CRISPR spacers reveals sampling biases and provides mechanistic insights in immune function" could and perhaps should be considered. As a side note, but an important technical aspect, PAM sequences and their predictions are independent from spacer sharing, and mechanistically agnostic to strand biases, so even experts could find this somewhat confusing.

We thank the reviewer for expressing an overall interest in our work and a thorough reading of the manuscript. Below we have addressed each point with great care.

We have altered the title to address your concerns to: "Comprehensive CRISPR spacer analysis reveals PAM-repeat associations and spacer selection preferences in single and co-occurring CRISPR systems". We have further removed spacer sharing to compatible arrays as it has yet to be demonstrated whether spacer sharing occurs.

Though many quantitative details and numbers are provided, I think some key aspects are missing and the authors must provide some information on materials and methods, as well as results, to provide a more thorough assessment of the work and interpretation of the data. With 200k spacers total and 70k encompassed in the study results, it is important to understand how many of these were perfect matches, or almost so (only 1 SNP) and how many were "distant matches". Providing a bar graph showing the numbers of spacers vs. level of identity or #/% of mismatches (SNPs and gaps) is likely necessary and could enable the authors to dig deeper in the best matches to have even more conclusive interpretations.

We thank the reviewer for this suggestion and provide the requested bar graph with number of spacers vs # mismatches (Supplementary Figure 2A) We further refer to this in the text in line 132-135: "For more than 25,000 of these, the best hit was completely identical to the spacer and for the vast majority (60,294) the total number of mismatched nucleotides was three or less (Supplementary Figure 2A)." We also have included a panel in that supplementary figure demonstrating the number of hits per spacer, where in case of multiple flanks per spacer, a consensus flanking region was generated. Using a consensus sequence compared to only the flanking sequence from a single hit further improved our PAM prediction (Supplementary Figure 2B).

On that note, in some cases, the authors seem to provide less "supported" arguments that focus on very specific and small subsets of the data (e.g. only 114 PAMs deciphered from such a large dataset seems implausible and much of the text is occasionally focused on a small subset of hand-picked cases (see later comment).

Please see our response to the next comment

With regards to materials and methods, the authors should discuss length and percent identity requirements and cutoffs for the blast hits and filtering and consider a deeper analyses of the best hits. With over 70k unique spacers making the cut, there should surely be some intriguing findings to glean from the set with perfect or near-perfect matches. I also imagine that there is a correlation between match quality and the ability to predict PAMs, which the authors should investigate. I also wonder what the distribution of these perfect matches is (e.g. supplementary figure 1) and what the proportion of perfect matches to prokaryotic viruses and also plasmids are.

We have more closely investigated the fraction of spacers with perfect matches and provided the requested supplementary figure demonstrating the distribution of perfect matches (Supplementary Figure 2C-2D).

We have also investigated the relationship between the match quality and the ability to predict PAMs (Supplementary Figure XB). We find that the signal-noise goes down with stricter criteria as there are less spacers with a hit per PAM and less hits per spacer (to build consensus flanking sequence) which reduces the PAM predictive power.

With respect to the last request (proportion prokaryotic viruses and plasmids) we believe that that is an interesting question, but warrants a separate study of its own, as identification of these from metagenomic sequences requires careful use of tools and proper validity checks and would deviate from the insights that we would like to present in this work.

As the authors know, some studies have investigated in depth naïve adaptation and primed adaptation in select organisms with known and well-characterized CRISPR-Cas systems (e.g. Marraffini, Severinov, Barrangou, Moineau and others). In cases where results are noteworthy (e.g. strand biases), one wonders whether these studies confirm and support these conclusions. To a similar extent, this applies to recent studies providing large scale analyses of the CRISPR spacers to the virome (e.g. Dion et al., 2021; Camarillo-Guerrero et al., and perhaps others, presenting highly relevant tools and data). This also applies to the relationship between CRISPR repeats and PAM sequences, with a need for the authors to determine whether this is indeed the case in a number of well-documented CRISPR-Cas systems with known PAMs. In many instances, the authors should ensure they properly discuss the highly relevant literature. It can admittedly be challenging to keep up with the CRISPR literature, but some of these references are highly relevant to provide context here.

We have added the requested literature references throughout the text and added more discussion in the relevant places for certain references.

In some cases, the authors show subsets of data that seems highly relevant and interesting (e.g. high matches for *Streptococcus*, *Pseudomonas* and *Staphylococcus*), and later on data with high levels of spacers with hits in select genera (see figure 1E, with perhaps the top 4-5 genera warranting deep analyses). Perhaps these should be analyzed a bit more in depth and given at least as much attention as select examples that otherwise stand out. In other cases, the authors seem to focus on

relatively small numbers of cases to make strong conclusions (e.g. 7 genomes, 17 genomes and 85 genomes for the three co-occurring systems compatibility discussion).

We believe this is an interesting subset to analyze in more depth. However, we believe this analysis will further detract from the main story line of the paper and will therefore make the story harder to follow, which is something the reviewer has indicated as a main concern. We have now included in the supplementary a table with the best hits for each spacer, which warrants further exploration for another study.

The 114 PAMs covering approximately 55% of spacers is intriguing and seems like a relatively low number of PAMs for such a large number of systems and spacers. Can the authors determine how many of these PAMs are known and already characterized (perhaps half of them have already been characterized and possibly most of them already predicted?).

We agree with the reviewer that the number of PAMs is relatively low and since most of our spacer hits come from Type I systems (80%), where we only find 43 PAMs this finding is even more striking. We suspect that the overall diversity in PAMs in Type I systems is generally low, as we find many clusters from many genomes using the same PAM (especially I-C, I-E and I-F). In the previous version we might have misrepresented our findings in the fact that we claimed to have found a large number of PAMs, whereas the more important result is that we have found it for a large number of spacers. Since we reperformed the analysis of Type II PAMs (see below), the overall number has changed and we now emphasize in the text that the PAM diversity differs greatly between Type I and Type II systems. Also an important feature of our work is not only finding new PAMs but also collecting which PAMs belong to which systems, so that a PAM can quickly be assigned to newly discovered systems or newly studied organisms.

We have to the best of our knowledge compiled a list of already pre-existing PAM sequences for Type I systems and added references in supplementary table 1 to those that have already been described. For Type II systems, as we now only determine PAMs for these systems on a genome basis, we cannot compare many to previously described PAMs in literature, but the ones we do match our predictions.

The comments and discussion about the critical use of the PAM for CRISPR array annotation and orientation is important. The authors should use this to discuss more extensively the quantitative extent to which previous studies have mis-predicted the orientation of CRISPR arrays.

Thank you for highlighting the importance of this section of the manuscript. We have modified the text to further discuss our findings that we got when comparing our predictions with experimental validation (TOP)

Another critical and intriguing finding is the presence of PAMs in 43% of Type III systems, which is in disagreement with mechanistic insights and the literature and could prompt a series of studies by the authors and perhaps others. This may need more discussion, highlighting and perhaps discussing specific type III systems that should be subjected to experimental analyses. Since cas genes working in trans has been shown before (e.g. with split arrays that share repeats and machinery), it is not unprecedented nor unbelievable that some Cas machinery could be shared between CRISPR-Cas systems, but the implications for DNA-targeting type I systems and RNA-targeting type III systems is

quite noteworthy and warrants additional coverage in my opinion. I actually wonder whether the authors could determine how often (or not) spacers are shared between these systems, and whether there are patterns of "activity" that could be investigated (are the array sizes similar reflecting "equivalent" evolutionary acquisition patterns between the two loci sharing acquisition machinery, or are there noteworthy differences that seem to indicate that the acquisition machinery is favoring one locus/system/CRISPR array, and is it in cis or in trans in these cases?). Perhaps some deep dive into the systems discuss in figure 5 is warranted.

We have created a new file with all genomes that contain a PAM (Supplementary File 4). This will allow genomes containing a Type III system with a PAM for experimental analyses. We have further guided the reader to this file in lines 261-262. We invite experimentalists to study whether these spacers are actually shared, while we report that at least from our bio-informatic predictions these co-occurring systems are compatible in both orientation and PAM preferences.

While the targeting strand bias is noteworthy, and has been discussed in the past in the literature, the authors must more clearly state that this is a biased and not a binary dataset. There are indeed biases (e.g 55%, 63%, 65%, ratios of approximately 2:1) and they do seem convincingly statistically significant, but they do not seem to have been selected against and there are explanation (e.g. steric access limitations due to other molecular mechanisms such as transcription) that could explain biases. Perhaps the authors could use transcriptomic data to determine whether transcription level correlates (or not) with biases.

We have further expanded the possible reasons for the observed biases including replication. Given the complexity of disentangling the causes of these biases and to keep focus in this manuscript we believe it is more suited for future bioinformatic and experimental studies.

Editorial comments:

- In the context of a system, the authors should and perhaps must use CRISPR-Cas (CRISPR and cas genes, and CRISPR and Cas proteins), as opposed to just "CRISPR", which refers to the repeat-spacer array. CRISPR itself and without Cas/cas is not a system...

We thank the reviewer for pointing this out and have replaced all instances of CRISPR systems to CRISPR-Cas systems

- In some cases, especially when discussing the original PAM discovery and naming convention, the authors should and perhaps must cite the 2008 papers by Deveau et al. and Horvath et al. which observed and documented and showed the actual PAM. The preceding Bolotin et al. 2005 paper did note a bias but predicted a perplexingly incorrect PAM and the subsequent 2009 Mojica paper renamed a previously established CRISPR motif. Given how familiar some of the authors are with the historical literature, it could be useful and valuable to the readership to cite some of the original literature (2007-2010), though the more recently published reviews cited are certainly defensible. This also applies to some recently published and highly relevant papers (self vs non-self; CRISPR

array identification tools, PAM identification and visualization tools and perhaps more like the two aforementioned virome-CRISPR spacer matching studies).

We thank the reviewer for suggesting to include more historical and recent literature to further link our results to other findings and studies and have included more references on the suggested topics.

- The authors should consider using the original verbiage and words used by the authors (e.g. CRISPRdb, CRISPRCasdb with the lower case database). In some instances, the authors use CRISPRCasDB, CRISPRCasDb and perhaps other variants.

We thank the reviewer for pointing this out and have replaced all instances of CRISPRCasDB to CRISPRCasdb

- In some cases, the authors should consider toning down some quantitative claims (e.g. 114 PAMs with perhaps as many already documented in the literature is not a "vast catalog", nor a "large number"

We have removed instances of these quantitative claims

Reviewer #2: In this work, Vink and coworkers perform extensive bioinformatics analyses on CRISPR spacers, gaining insights into subtype-specific PAM preferences, biases in target strand selection, and potential crosstalk between co-occurring systems. Multiple studies have analyzed spacer sequences broadly, although the limited number of matches have restrained any major insights. The authors report an incredible increase in matches—presumably due to the increasing accumulation of metagenomic sequences, allowing them to garner these insights. While many of the insights the authors draw are known and understood in the field (e.g. subtype-specific PAM preferences, PAMs and repeats being strongly different), the authors do provide additional insights as well as a large-scale analysis that confirms what the field had observed anecdotally. My comments noted below should strengthen the work and make it a stronger resource for the CRISPR community.

We thank the reviewer for the interest in our work and for suggestions to improve the manuscript.

Major comments:

1. Because the work relies solely on bioinformatics analyses, some of the conclusions need to be softened or expanded with other explanations. As one example, on line 335 - 337, the authors state that the co-occurring systems may use each other's spacers. However, a competing explanation is that the acquisition machinery recognizes both repeats, but differences in the repeats prevents spacer sharing. Providing experimental evidence would be even stronger, although I recognize this would add considerable work.

We appreciate the concerns with our hypotheses and have expanded our discussion of possible explanations for our results in the discussion. We hope that our work can inspire further experimental work to be done based on it, but believe it is out of scope to do so in the current study.

2. The identified PAMs reflect requirements for spacer acquisition, while the requirements for

targeting can deviate strongly and tend to be much more relaxed (e.g. acceptance of PAMs by *E. coli* Cascade outside of AAG). However, the authors often focus on targeting when assessing PAMs. This distinction between acquisition and targeting requirements could be clearer throughout, and I recommend that the authors start from the perspective of acquisition sequences that lead to targeting.

We agree that a large part of the observed motifs in the selected spacers in nature are based on the acquisition process. However different from experimental acquisition assays, the spacers that we find are furthermore likely selected by natural selection of effective spacers in providing an interference response. PAMs are therefore a combination of SAM (spacer acquisition motif) and TIM (target interference motif) (Shah et al., 2013), although these terms never became mainstream. We have added a new paragraph in the introduction to discuss this in lines 57-72.

3. PAMs have been experimentally determined for a large number of Type II systems (e.g. Gasiunas et al. Nat Comm 2020). Incorporating these known PAMs into some of the analyses would provide more experimental support. It is also worth noting that PAMs can vary between highly similar Cas9 nucleases with similar repeats, complicating analysis of PAMs associated with clusters.

We have introduced a new analysis in our manuscript as the result of this comment (and minor comment 11). We already hypothesized that our analysis with clusters would in some cases result in complications when analyzing Type II PAMs, but when we analyzed this in more depth we found for many clusters divergent PAMs within the same cluster. We have repeated this analysis for Type I and Type III systems and find no such effect (Supplementary Figure 5). We have therefore concluded that for Type II systems the PAM have to be determined on an individual basis and repeat clusters cannot be used. We have compared several of the PAMs that we find for individual genomes with experimental results and get a good agreement. We have modified the text in lines 226-239 to explain these results.

4. Could the direction of replication affect the bias in protospacer orientation? Such biases are known in bacterial genomes. If they exist in viral or plasmid genomes as well, then that could explain biases—particularly based on the typical source of spacers from one subtype to the next. This bias could upend the authors' assertions about preferring or avoiding RNA targets.

We thank the reviewer for the suggestion and have added this as a possibility in our discussion of this topic: “Lastly, we cannot exclude the possibility that DNA replication might cause the observed strand bias for some subtypes, as transcription and replication are often co-oriented in prokaryotes, plasmids and phages (Brewer, 1988; Srivatsan et al., 2010).”

Minor comments:

5. The writing was generally clear, although there were typos and grammatical errors scattered throughout the text that were distracting.

We have further improved the manuscript by correcting typos and grammatical errors.

6. Line 36: I would recommend "CRISPR immunity" rather than "CRISPR interference", as the latter is commonly understood to mean programmable gene repression with CRISPR.

CRISPR immunity refers to the whole defense pathway whereas we refer to a particular stage in the immune process, namely interference, which is the most commonly used nomenclature in literature (van der Oost et al., 2009). We therefore did not alter this.

7. Line 59: a much more recent review on alternative functions was by Wimmer & Beisel, Front Microbiol 2020.

We thank the reviewer for this recommendation and added the reference to the text.

8. Line 77 - 78: This is an overstatement and should consider the use of cell-free transcription-translation systems (e.g. Marshall et al. Mol Cell 2018) that are easier than the IVT system cited by the author. Instead, the valid argument is that experimental systems require testing one system at a time that can't cover the same space as bioinformatics approaches.

We have modified the text to the following: "High-throughput assays to identify the PAM of CRISPR-Cas systems have been developed, but remain limited compared to the total range of CRISPR-Cas systems accessible with bio-informatic tools ." We have also added the reference.

9. Line 85 - 87: the logic seems circular here, where PAM predictions were used to improve PAM predictions.

Thank you for identifying this error. It has now been removed.

10. Line 98: better define what is meant by a match. Is this a perfect match? Or with some mismatches? This is likely covered in the methods, although this should be noted here given that identifying spacer matches is the root of the entire work. Also, does their definition help explain why the authors find many more matches than prior attempts at spacer matching?

We have added a supplementary figure which demonstrates the distribution of mismatches over the hits we get. As our filtering approach is complex we do not include the explanation in the main text. We believe our filtering approach does significantly add to the number of matches found, but that the further addition and expansion of big metagenomic datasets are also responsible for this.

11. Line 196: is there evidence of distinct sets of PAMs within these clusters? This is worth closer inspection.

Yes, please see our response to major comment 3, where this topic is also addressed.

12. Line 197: add Dugar et al. Mol Cell 2018 and Rousseau et al. Mol Cell 2018, which also showed RNA targeting by the CjeCas9 and NmeCas9 without any PAM requirements.

We have added the references to the text

13. L. 184: this is a bold claim, especially since none of the provided examples seem particularly

novel compared to what's been experimentally determined (particularly for Cas9 nucleases).

We have added to the best of our knowledge previously described PAMs for Type I systems and indicated which PAMs are novel in this manuscript. For Type II systems, this is harder to compare because the PAMs are less well defined.

14. Line 260 - 300: while the authors do observe significant biases based on binomial distributions, the differences are often minor by eye. These smaller differences should be reflected in the text, such as by providing actual percentages throughout.

We have added percentages where they were missing.

15. Line 318 - 319: It's not obvious at this point in the text has any preferences would enable or preclude compatibility. Perhaps add another sentence or two.

We have added the following sentence to explain our line of thought: The subtype-specific preference of template targeting (e.g. in Type I-E and Type II) will reduce the number of effective spacers that can be used by co-occurring RNA-targeting systems, whereas subtypes with a preference for the coding strand (Type I-A, Type I-B) might make their spacers more compatible for RNA-targeting systems.

16. Line 329 - 331: spacer sharing would be extremely unlikely given recognition of a hairpin and handle for the I-B system and pairing with the tracrRNA for the II-A system. The more plausible explanation is that these systems recognize common PAMs that happen to be the complement of each other.

We agree with this statement and to shorten this section this part has been removed from the text.

17. Line 345: a nice experimental example to work in the story was an example of one set of acquisition machinery incorporating spacers into a co-occurring II-C and VI-B system (Hoikkala et al. mBio 2021).

Indeed. We had already included this reference somewhere else, but now also include it in this part of the story.

18. Line 378 - 381: Or could the I-C system recognize an A-rich sequence on the opposite strand? Checking available crystal structures could address this possibility.

This is an interesting possibility. We have checked the crystal structures and can confirm that the I-C system recognizes 5'GAA3' overhangs and the I-E system recognizes 5'CTT3' overhangs. They therefore both probe the same strand but recognize complementary nucleotides. We have added this to the text in Lines 420-424.

19. Line 407 - 410: This sentence could be shortened or split in two.

We thank the reviewer for this suggestion and have split the sentence.

Second round of review

Reviewer 1

Vink et al. present a revised version of their study of CRISPR spacer diversity analysis.

The updated title is noted, though it seems to be un-necessarily complex and long, and it also should refer to “CRISPR-Cas” in the context of “systems”, or if just “CRISPR” is selected, then it should be used in the context of “arrays”. The latter may forego the need to use “CRISPR” twice in the title.

The added data is also noted and benefits the readership, for instance with regards to the number and type of spacer:protospacer matches and the PAMs. Perhaps some of the supplemental material should be considered for inclusion in the main portion of the manuscript, but this is up to the authors and editors.

The discussion and issue raised / corresponding rebuttal about PAM paucity, notably as it applies to Type I systems is interesting, and I note the indeed striking identification of only 43PAMs, including “many clusters from many genomes using the same PAM (especially I-C, I-E and I-F)”. The new focus on PAM diversity in general and Type I vs. Type II in particular is interesting and derived from this exchange and I think some of the supplemental material would warrant inclusion in the main text for this topic.

I do think that for PAMs that have been characterized in the literature and especially for PAMs characterized from canonical (per Makarova et al.) and model systems (well documented functionally), it would be proper and perhaps even necessary to specify which species (and perhaps even strains if applicable) are in play (e.g. for figure 2 and perhaps figure 4, as well as Supplementary Table 1). Well known models for Type I (especially I-E and I-F) and type II (some II-A and perhaps a II-C) should be in play.

With a connection between CRISPR repeat sequence and PAM sequence made, I think the authors should quantify these and show extent of diversity quantitatively and determine whether there are differences that are statistically significant (or not) between Types I and II. I also think it would be proper to delve more into further supporting some of these claims by calling out relevant functional data from model systems (see and cross-connect the literature and Figure 2 and supplemental table 1).

Reviewer 2

The authors have taken reasonable steps to address all comments from the reviewers. I have no other comments for this work.

Authors Response

Point-by-point responses to the reviewers' comments:

Reviewer #1: Vink et al. present a revised version of their study of CRISPR spacer diversity analysis.

We thank the reviewer for a thorough second round of review and appreciate the positive reception of our added data.

The updated title is noted, though it seems to be un-necessarily complex and long, and it also should refer to “CRISPR-Cas” in the context of “systems”, or if just “CRISPR” is selected, then it should be used in the context of “arrays”. The latter may forego the need to use “CRISPR” twice in the title.

We further adapted and shortened the title to overcome the points raised by the reviewer to “PAM-repeat associations and spacer selection preferences in single and co-occurring CRISPR-Cas systems”

The added data is also noted and benefits the readership, for instance with regards to the number and type of spacer:protospacer matches and the PAMs. Perhaps some of the supplemental material should be considered for inclusion in the main portion of the manuscript, but this is up to the authors and editors.

The discussion and issue raised / corresponding rebuttal about PAM paucity, notably as it applies to Type I systems is interesting, and I note the indeed striking identification of only 43PAMs, including “many clusters from many genomes using the same PAM (especially I-C, I-E and I-F)”. The new focus on PAM diversity in general and Type I vs. Type II in particular is interesting and derived from this exchange and I think some of the supplemental material would warrant inclusion in the main text for this topic.

Based on the above comments, we have included parts of supplementary figure 2 and supplementary table 1 into the main text.

I do think that for PAMs that have been characterized in the literature and especially for PAMs characterized from canonical (per Makarova et al.) and model systems (well documented functionally), it would be proper and perhaps even necessary to specify which species (and perhaps even strains if applicable) are in play (e.g. for figure 2 and perhaps figure 4, as well as Supplementary Table 1). Well known models for Type I (especially I-E and I-F) and type II (some II-A and perhaps a II-C) should be in play.

In all these figures, to provide complete information (all species/repeat sequences) would mean the figure/table would get very large or unreadable. We have therefore only given a representative genus in Figure 2 and supplementary table 1 even though the PAM (logo) is determined from a cluster of repeat sequences found in multiple genomes/species. To find which

species contain a certain PAM, we would refer to Supplementary file 4 (now named additional file 5). This file contains the accession number for every species for which we can determine a PAM. Since this list is almost 5000 entries long we cannot include it in the main text. We have more strongly emphasized readers to look into these files in all captions of the figures

In our comparisons to models in Supplementary Table 1, we have only included those models which contain repeat sequences that fall within one of the repeat clusters for which we have determined a PAM. We therefore have good validation of our method with these canonical systems. We now mention this in the caption of this figure.

In Figure 4, where we investigate strand bias, we added all spacers of all organisms belonging to the genera *Moraxella* and *Escherichia* to be able to produce the charts. For those interested in specific strains, this could be extracted from additional file 2. However due to the sheer amount of data this again cannot be depicted in a figure or table.

With a connection between CRISPR repeat sequence and PAM sequence made, I think the authors should quantify these and show extent of diversity quantitatively and determine whether there are differences that are statistically significant (or not) between Types I and II. I also think it would be proper to delve more into further supporting some of these claims by calling out relevant functional data from model systems (see and cross-connect the literature and Figure 2 and supplemental table 1).

We have further improved this section by visualizing the PAM diversity with respect to the number of repeat clusters and the number of genera (included in Supplementary Figure 5E-F). We do not think there is a suitable statistical analysis for this analysis, given the non-random distribution of spacers within each CRISPR-Cas subtype each drawn from a different set of phylogenetic taxa (e.g. some subtypes are drawn from more homogeneous set of organisms than others), however given the great differences between Type I and Type II systems we believe our conclusions are justified. We further discuss this point in the main text in the paragraph starting with the following sentence: “Overall, the diversity in PAM motifs in Type II systems is higher than in Type I systems.”

For supplementary table 1 we have found 15 studies in which experimental PAMs match our predictions. We have added the studies that match to data in Figure 2 as well in its caption. We further looked for experimental PAMs studies that match predicted PAMs, but were unable to retrieve more relevant PAMs than the ones we had already found. We have also added multiple references that link our results to previous findings, discuss PAM diversity and CRISPR cooperativity in the results, discussion and caption sections.

Reviewer #2: The authors have taken reasonable steps to address all comments from the reviewers. I have no other comments for this work.-